# Life-history adaptation under climate warming magnifies the agricultural footprint of a cosmopolitan insect pest

Estelle Burc[1,2,7], Camille Girard-Tercieux[1,3,4,7], Moa Metz[1,5], Elise Cazaux[1,3], Julian Baur [1], Mareike Koppik[1,6], Alexandre Rêgo[1], Alex F Hart[1] & David Berger [1] ✉

Climate change is affecting population growth rates of ectothermic pests with potentially dire consequences for agriculture and global food security. However, current projection models of pest impact typically overlook the potential for rapid genetic adaptation, making current forecasts uncertain. Here, we predict how climate change adaptation in life-history traits of insect pests affects their growth rates and impact on agricultural yields by unifying thermodynamics with classic theory on resource acquisition and allocation trade-offs between foraging, reproduction, and maintenance. Our model predicts that warming temperatures will favour resource allocation towards maintenance coupled with increased resource acquisition through larval foraging, and the evolution of this life-history strategy results in both increased population growth rates and per capita host consumption, causing a double-blow on agricultural yields. We find support for these predictions by studying thermal adaptation in life-history traits and gene expression in the wide-spread insect pest, *Callosobruchus maculatus*; with 5 years of evolution under experimental warming causing an almost two-fold increase in its predicted agricultural footprint. These results show that pest adaptation can offset current projections of agricultural impact and emphasize the need for integrating a mechanistic understanding of life-history evolution into forecasts of pest impact under climate change.

Climate change is predicted to impact growth rates and distribution ranges of ectothermic pests[1–7] with potentially severe outcomes for agricultural economics and world food supply[8–12]. To mitigate these events, forecasting of pests' responses to future climates is an important task for biologists[13,14]. Current forecasts use deterministic models that rely on information about species niches and climate data. However, these projections typically do not incorporate the potential

for genetic evolution of climate niches and key life-history traits that mediate the effect that pests have on their hosts[13,14]. Indeed, the evolution of ecological niches has been rapid in many cosmopolitan pests and not accounting for this adaptive potential may therefore provide inaccurate forecasts of their future agricultural impact[1,7,13–17].

Ectothermic organisms are constrained by temperature-dependent biophysical properties of cells[18–22]. On the one hand,

[1]Department of Ecology and Genetics, Program of Animal Ecology. Uppsala University, Norbyvägen 18D, 75236 Uppsala, Sweden. [2]Agronomy Institute Rennes-Angers (IARA), Graduate school of agronomy, 35000 Rennes, France. [3]Université de Toulouse, Toulouse INP-ENSAT, 31326 Castanet-Tolosan, France. [4]Université de Lorraine, AgroParisTech, INRAE, UMR Silva, 54000 Nancy, France. [5]Department of Biology, Faculty of Natural Sciences, Norwegian University of Science and Technology, 7491 Trondheim, Norway. [6]Department of Zoology, Animal Ecology, Martin-Luther University Halle-Wittenberg, Halle (Saale), Germany. [7]These authors contributed equally: Estelle Burc, Camille Girard-Tercieux. ✉e-mail: david.berger@ebc.uu.se

metabolism is constrained by cold temperature that slows down both juvenile development and adult reproduction[19], predicting that climate warming may lead to faster population growth rates via plastic increases in these rate-dependent life-history traits[23,24] and thereby worsen pest impact in many parts of the world[2]. On the other hand, acute hot temperatures can jeopardize cellular homoeostasis[25–27], leading to rapid fertility decline[28,29] and increased mortality[22,30], which predicts that climate warming may reduce insect growth rates in the world's hottest regions[2,31]. These thermodynamic constraints are thus expected to have a strong influence on ectotherm pests facing climate warming, but how genetic adaptation to temperature may influence long-term outcomes remains unclear. For example, theory predicts that thermal adaptation should be governed by trade-offs and that shifts in physiology allowing superior performance at one temperature extreme should lead to reduced performance at the other[18,20,32,33]. However, when thermal adaptation occurs it often does so seemingly without any apparent trade-offs, with some genotypes displaying superior performance across the entire thermal range studied[19,33–35].

Indeed, a typical signature of insect pests is their broad climate niches coupled with fast growth rates[1,16]. One mechanism that can allow organisms to occupy broad climate niches is upregulation of molecular chaperones that aid cellular homoeostasis under harsh conditions[20,36]. Such responses are energetically costly and therefore need to be compensated by either allocation of resources away from reproduction in favour of cellular maintenance[20,36], or by acquiring more resources overall by increased food intake[5,19]. Notably, the first strategy, involving allocation trade-offs, is predicted to increase survival but reduce insect reproductive rates at stressful temperatures, whereas the second strategy, to increase resource acquisition, permits both sustained adult survival and reproductive output but is only possible if larvae can increase their host consumption. This makes the prediction that the effect of climate warming on insect growth rates may critically depend on the access to host plants[37]. Moreover, applying this reasoning to pest species suggests that the rich host abundance supplied by agricultural crops might make the strategy to increase energy acquisition particularly favourable, which could result in increased per capita host consumption by pests facing stressful temperatures. These different routes to deal with stressful temperatures can thus have important consequences for predicting the agricultural impact of insects under climate change[1,14,17,38–40].

Despite its importance, however, how insect life-histories are optimized under climate change remains unclear[1,16,39]. Moreover, for adaptation to occur, genetic variation is a prerequisite. Geographically isolated populations can differ greatly in the abundance of both segregating and fixed genetic variation due to differences in their evolutionary past, which may affect future adaptive potential[41–45]. Such historical contingencies may be of particular importance in pests species, as geographically isolated populations often originate from human-mediated long distance dispersal events of only a few individual genetic founders. Evolution under climate change may therefore proceed along different trajectories and rates across a species distribution range, even when climates change in parallel.

Here we explore how thermal adaptation in life-history traits of insect pests affect their population growth rates and host consumption. We first construct a model of thermal adaptation in insect life-history by combining theory on thermodynamics[46,47] and acquisition and allocation trade-offs[48,49]. The model predicts that warming temperatures not only increase the potential for fast growth rates, but also benefit the evolution of increased host consumption in larvae, which allows sustained survival and reproduction in adults, causing a double-blow to agricultural yields. We then employ experimental evolution in the seed beetle, *Callosobruchus maculatus*, an economically important cosmopolitan pest on legume seeds[50]. Following 5 years of experimental evolution at either hot or cold temperature, we assay several life-history traits predicted to affect agricultural loss, including

development rate, metabolic rate, body size and adult offspring production. In accordance with model predictions, we find that life-history evolution in response to hot temperature causes an almost two-fold increase in the agricultural footprint of *C. maculatus*. This increase is mediated by increases in larval host consumption, in turn leading to increased adult body mass and reproductive output. This effect of life-history adaptation is contingent on the genetic make-up of the founding populations and is not observed in populations adapting to cold temperature. These findings suggest that equipping species distribution models with genetically explicit details accounting for life-history evolution can lead to a substantial improvement in predictive accuracy of pest impact under climate change, even over relatively short time scales.

## Results
### The agricultural impact of thermal adaptation via resource acquisition and allocation

To formalize predictions for how thermal adaptation in insect life-history will impact agriculture, we combined the classic theoretical framework of resource acquisition and allocation trade-offs[48,49,51,52] with the thermodynamic laws that govern enzyme kinetics[20,53]. Warm temperatures increase productivity rates of ectotherms[19,23] because biological rates are at some level governed by enzymatic reaction rates, $k_B$, that scale with temperature according to[46,47]:

$$k_B \propto A_B e^{-Ea_B/RT} \tag{1}$$

where $Ea_B$ is the activation energy required for the reaction to occur, $R$ is the universal gas constant, $T$ is temperature in Kelvin, and $A_B$ is a species and rate specific constant that may itself evolve to accommodate effects of temperature[23,54,55] (Fig. 1A). Complementing Eq. (1) with the power law describing how productivity scales with body mass, $m$, results in the well-known prediction of mass and temperature-dependent population growth based on the metabolic theory of ecology[23,24]: $r \propto m^b k_B$, where $b$ is the allometric exponent, often taken to be 0.75 based on the physics of distribution networks in plants and animals[23,56] (but see refs. [57–59]).

However, excessively hot temperatures lead to increased molecular failure rates[30] due to a range of detrimental processes that also scale with temperature, including protein unfolding[25,26], metabolic expenditure[37,60], production of reactive oxygen species[20], and oxygen depletion[61]. Several studies have highlighted that activation energies ($Ea$) can themselves be temperature-dependent, which is particularly apparent for molecular failure rates, that show a disproportionate increase at hot temperatures[18–20,30,62] (Fig. 1A). To account for this increase, mortality rate can be described by Eq. (1) with its own, temperature-dependent, activation energy, $Ea_D(T)$[21,26,30,63,64] and rate-specific constant ($A_D$):

$$k_D \propto A_D e^{-Ea_D(T)/RT} \tag{2}$$

As follows, population growth rate is the result of temperature-dependent productivity and mortality rate[1,2] (Fig. 1B). Inherent enzyme properties governing the temperature-dependence of reaction rate kinetics have been argued to evolve more slowly than thermal adaptation via evolution of compensatory stress responses[20,36,55,65–67]. Such compensatory responses involve upregulation of molecular chaperones, such as heat-shock proteins[20,68,69], that maintain cellular homoeostasis at stressful temperatures but are costly to produce[36,68,70]. Here we therefore integrated the thermodynamic predictions of Eqs. (1) and (2) into a life-history framework incorporating energy acquisition and allocation trade-offs[48,49,70]. First, assume that energy reserves, $M$, are accumulated by larval host feeding at a cost, $c$, so that survival to maturity equals: $(1-c)^M$. Further assume that accumulated energy reserves can be allocated with fraction $p$ to increase enzymatic

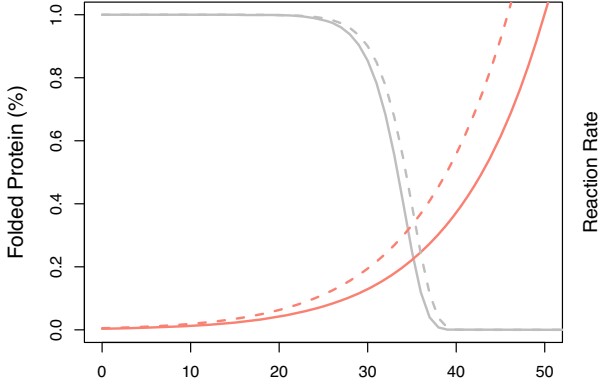

**A: Effects of energy acquisition on enzyme performance**

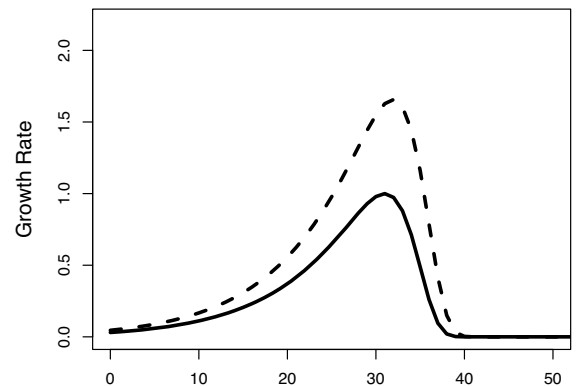

**B: Effects of increased energy acquisition on growth rate**

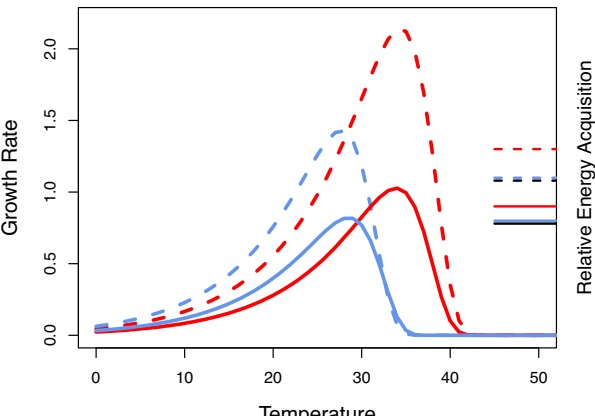

**C: Thermal adaptation via increased energy acquisition**

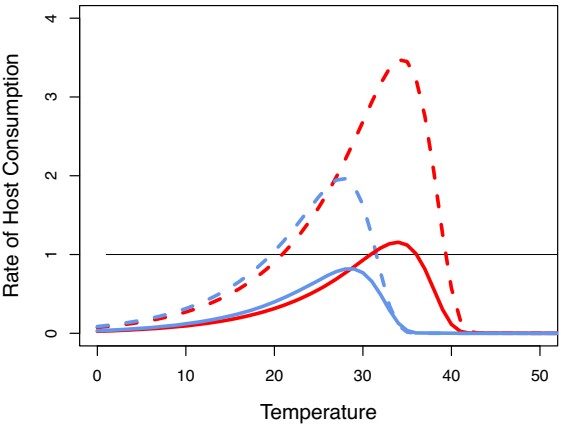

**D: Predicted Agricultural Impact**

**Fig. 1 | Thermal adaptation via compensatory energy acquisition can increase pest species' impact on agriculture.** Shown are predictions based on protein kinetics (details in Supplementary 1). **A** Temperature increases enzymatic reaction rates[20,56] (pink lines) while too hot temperatures lead to (reversable) protein unfolding (grey lines), leaving fewer enzymes available to do work[25,26]. Reduced costs of feeding (broken lines) leads to more energy reserves acquired ($M$), of which a fraction ($p$) can be devoted to increased production of catalytic enzymes that speed up constrained reaction rates at cold temperature[20], and another fraction ($q$) that can be devoted to molecular chaperones that stabilize proteins at hot temperature[36]. **B** The resulting growth rate is the product of the remaining reserves that can be devoted to growth and reproduction, $M(1-p-q)$, the reaction rate, and the proportion of properly folded protein. **C** The resulting growth rates for populations adapting to cold (23 °C; blue) and hot (35 °C; red) temperature via compensatory responses (increased $M$, and allocation to $p$ and $q$) are shown for

high (full lines) and low (broken lines) costs of feeding. While the cold-adapted population outcompetes the hot-adapted population at cold temperature, growth rates are greater at warm temperatures and for the hot-adapted population due to thermodynamic constraints on reaction rates[23,56]. Horizontal lines on the right-hand y-axis show $M$ at the optimal strategy; lower costs of feeding favour increased energy acquisition, which improves thermal performance and increases thermal niche breadth. In the depicted scenario, increased energy acquisition is particularly beneficial at hot temperature that puts increased demands on cellular maintenance to ensure protein stability. **D** The possibility to evolve thermal performance via compensatory energy acquisition leads to an even greater agricultural impact (the product of $M$ and population growth rate; compare difference between full and hatched lines in (**C**, **D**)). All panels show responses relative to the maximum value for ancestors reared at benign 29 °C at high costs of feeding (black full line in (**B**).

reaction rates, and with fraction $q$ to decrease molecular failure rates, leaving $M(1-p-q)$ resources that can be devoted to productivity, $B$, (i.e. growth and reproduction) according to:

$$B \propto [M(1-p-q)]^b; \quad A_B \propto pM; \quad A_D \propto (qM)^{-1} \quad (3)$$

where $A_B$ and $A_D$ are the rate-specific constants for productivity and mortality, respectively. Note that we here chose to model effects of energy acquisition and allocation via $A$, and not $Ea$. This choice has both a practical and biological motivation as parametrization is straightforward and seems justified; a proportional change in allocation to $A$ leads to a proportional change in reaction rates, but we acknowledge that our approach does not capture the full range of possibilities for how energy allocation and acquisition trade-offs can

affect thermal performance[20,65,71–73]. Population growth rate, $r'$, is then:

$$r'(T) \propto (1-c)^M * B * A_B\, e^{-\frac{Ea_B}{RT}} * \frac{1}{1+A_D e^{-\frac{Ea_D(T)}{RT}}} \quad (4)$$

We can find the optimal strategy of juvenile resource acquisition (increasing $M$ at cost $c$) and allocation (of $M$ to $p$ and $q$) at different temperatures by numerically solving for the combination of $M$, $p$ and $q$ that maximize $r'$. The solution to this multidimensional trade-off depends on the specific form of Eq. (1–3) and parameters that govern temperature sensitivity (i.e. $Ea$, $A$ and their implementation); mechanistic details which are empirically not well-defined. Indeed, Eqs. (1–2) represent phenomenological rather than mechanistic

relationships, emphasizing the need for further experimental work quantifying these relationships.

Nevertheless, the qualitative predictions from this heuristic model are robust (see also ref. [37]). We provide a parameterized example in Fig. 1 grounded in knowledge of how temperature affects protein catalysis and stability[21,25,27,63,64,67]. Note that, because our focus is on climate warming, our example models the effects of temperature on reaction rates and the detrimental effects of acute hot temperatures via loss of protein stability, but does not model lethal effects of acute cold temperatures (details in Supplementary Note 1). First, and as pointed out previously[2], warm temperatures increase growth rates and agricultural impact via thermodynamic effects on enzyme reaction rates. Second, hot temperatures that jeopardize survival favour increased larval resource acquisition and more energy reserves ($M$) that can be allocated to maintenance[37] (Fig. 1B, Supplementary Note 1). When increased resource acquisition comes at low cost, this life-history strategy is particularly favourable and results in increased thermal niche breadth and higher population growth rates (Fig. 1C). Third, this has an even greater agricultural impact, given by the product of population growth rate ($r$) and per capita host consumption (assumed proportional to $M$) (Fig. 1D). In fact, the strategy to increase resource acquisition at hot temperature causes populations experiencing temperatures slightly higher than those which maximize growth rate to have the greatest agricultural impact (Supplementary Note 1).

## Experimental evolution of life-history traits and agricultural impact

To evaluate the prediction that trait and temperature-specific responses in pest species' life-history can influence their agricultural impact under climate change, we compared twelve experimental evolution lines of the seed beetle *C. maculatus*; a capital breeding holometabolous insect that when kept in aphagous conditions acquires all its resources for adult reproduction and survival in the larval stage[74]. Six lines were adapted to 35 °C (hot) and six lines adapted to 23 °C (cold). All lines were reared on seeds from their preferred and economically important host, black-eyed bean (*Vigna unguiculata*) (Fig. 2). The lines were derived from three geographically isolated founding populations (ancestors, kept at 29 °C) originating from Brazil, California (USA) and Yemen[75], allowing us to assess the importance of geographic differences and evolutionary history for future responses to climate change.

## Molecular evidence for a trade-off between reproduction and thermal tolerance

To provide evidence for the model assumption of a trade-off between investment in reproduction and cellular maintenance under thermal stress (equation 3), we analysed data on gene expression from two separate experiments on *C. maculatus*. These datasets contained genes that were significantly differentially expressed in female abdomens in response to heat shock, and between virgin and reproducing females. We identified 1269 reproduction-related genes and 765 heat-stress genes. We found 137 genes present in both datasets, which is significantly more than expected by chance (exp: 83.4 genes, $\chi^2 = 37.5$, df = 1, $p < 0.001$, Fig. 3A). Moreover, only 3 of the 137 genes showed concordant responses to mating and heat-shock (all three decreased in expression), whereas 134 genes showed antagonistic responses (binomial test: $P < 0.001$, Fig. 3B), providing strong evidence for a trade-off between reproduction and the heat-stress response. Out of these genes, 21 were upregulated in response to reproduction and downregulated in response to heat. These genes included common ribosomal genes that are crucial for protein translation, signifying reproductive investment (Supplementary Data 1). The 113 antagonistic genes that were downregulated in response to mating and upregulated

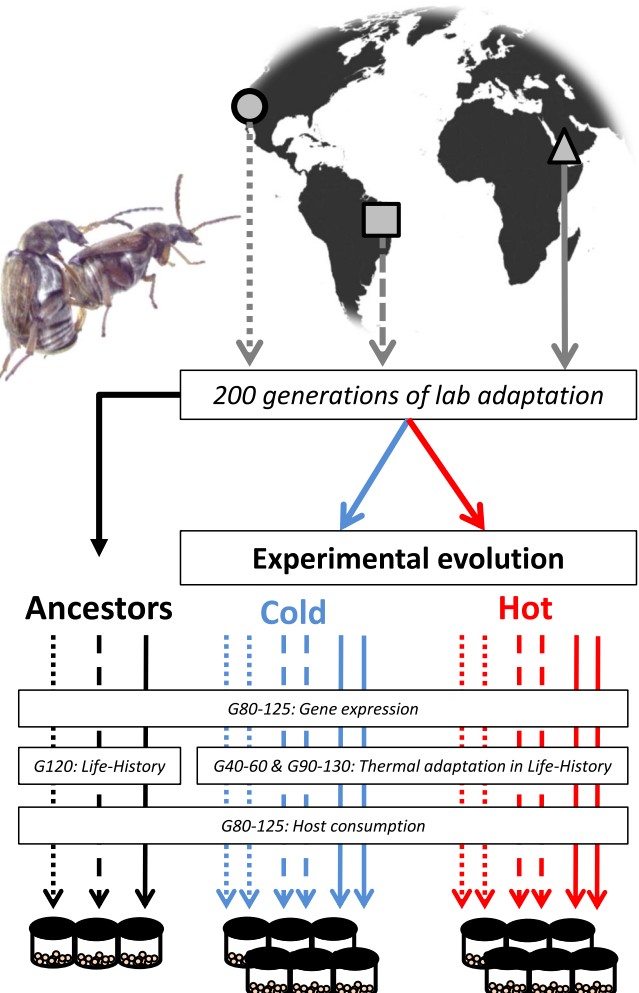

**Fig. 2 | Experimental evolution of thermal niches and life history in seed beetles.** Dryland legumes are nutritionally rich, drought tolerant and resilient to harsh weather[11,140] and provide subsistence for more than half a billion people in the driest regions of the world[11]. However, these crops suffer substantial losses from seed beetles[140]. To investigate how genetic adaptation to climate change might shape the impact of seed beetles on legume crops, we performed large-scale phenotyping of long-term experimental evolution lines of *C. maculatus*. Populations from three geographic locations (Brazil, square; California, circle; and Yemen, triangle) were sampled and allowed to adapt to common laboratory conditions for ca. 200 generations before experimental evolution. Four replicate populations were created per genetic background and split between the two evolution regimes (23 °C−cold, and 35 °C−hot), while the original ancestor remained at the benign temperature (29 °C). Experimental evolution proceeded over 90 and 130 generations for the cold (blue) and hot (red) adapted populations. During this time, the ancestors went through 130 generations of evolution. To provide molecular evidence for a temperature-dependent trade-off between maintenance and reproduction, gene expression was measured in both evolved and ancestral lines after 80−125 generations of evolution. After 40 (cold adapted) and 60 (hot adapted) generations, life-history adaptation was assessed in a common garden experiment. This data was complemented in generations 90 (cold lines) and 130 (hot lines and ancestors). Life-history traits were measured in the three ancestors after 120 generations of evolution. Following 80−125 generations of experimental evolution, the host consumption of all lines and ancestors was scored in a common garden to assess how life-history evolution under simulated climate change had shaped the agricultural footprint left by *C. maculatus*. Photo-credit seed beetles: Johanna Liljestrand Rönn. The world map was made using the R packages ggplot2[141] and ggthemes[142].

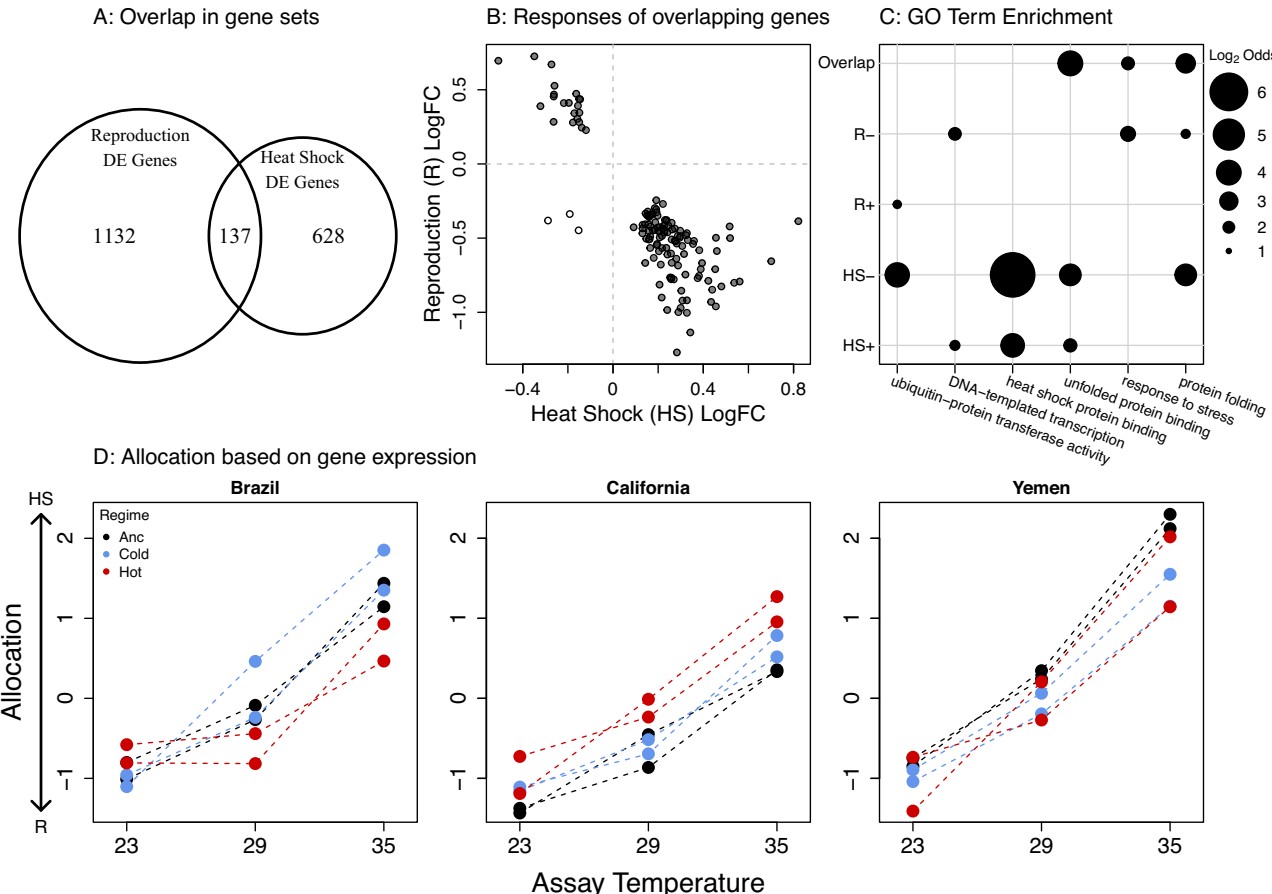

**Fig. 3 | Molecular evidence for a temperature-dependent trade-off between maintenance and reproduction. A** Differentially expressed genes in response to heat shock (HS) and mating (R); 137 genes responded to both treatments, which was more than expected by chance. **B** Of these genes, all but three showed antagonistic responses to the two treatments, signifying a trade-off between investment in reproduction and maintenance under heat stress. **C** GO term enrichments for genes upregulated (+) or downregulated (−) in response to mating (R) and heat stress (HS), as well as for the overlapping antagonistic genes. **D** The expression of the 134 antagonistic genes in the experimental evolution lines and ancestors indicate a clear increase in allocation towards cellular maintenance at the cost of reduced reproduction when beetles were reared at 35 °C (the thermal regime of hot-adapted lines). However, there were no consistent differences in expression between ancestors (black) and the cold (blue) or hot (red) evolution regime.

in response to heat stress included common insect heat-shock proteins (Supplementary Data 1). GO enrichment analysis showed that the 134 antagonistic genes that overlapped both gene sets were enriched for biological process involved in protein folding and degradation of unfolded proteins, which are essential components of cellular maintenance under thermal stress (Fig. 3C, Supplementary Data 2).

To confirm that the revealed trade-off also was at play in the thermal conditions used during experimental evolution, we analysed gene expression data from all cold and hot lines, as well as their ancestors, reared in a common garden design including 23, 29 and 35 °C assay temperature. Each evolution line was represented by a single mRNA library per temperature, based on a pool of 10 abdomens from mated young females of the same age as the females phenotyped for life-history traits (see further below). The ancestors were represented by two such replicates. To evaluate investment into reproduction versus maintenance, we calculated a score for each of these mRNA libraries along the axis describing the trade-off based on the expression of the 134 antagonistic genes (Fig. 3B). We then analysed differences between assay temperatures and evolution regimes using nested ANOVAS (Supplementary Table 1). Beetles reared at 35 °C showed a strong change in allocation towards cellular maintenance and away from reproduction compared to beetles reared at the other

two temperatures (assay temperature: $F_{2,17} = 263.0$, $P < 0.001$, Fig. 3D). There were no significant differences between evolution regimes overall ($F_{2,8} = 0.12$, $P = 0.89$), although the differences between the evolution regimes and their ancestors showed a dependence on geographic origin (interaction: $F_{4,8} = 11.0$, $P = 0.002$). Qualitatively similar results were obtained if restricting the analysis to the 113 genes that were upregulated in response to heat stress (lower right quadrant of Fig. 3B) or when analysing all 765 heat stress genes and 1269 reproduction genes (Supplementary Figs. 1 and 2, Supplementary Tables 1c–e). When restricting the analysis to the 21 overlapping genes that were upregulated in response to mating (upper left quadrant of Fig. 3B), however, ancestors showed overall higher expression than both the cold and hot regime (Supplementary Fig. 1, Supplementary Table 1b), hinting at a potential decrease in some aspects of reproduction early in life during experimental evolution. These analyses nevertheless imply that there has been relatively little temperature-specific adaptation in resource allocation overall (i.e. $q$ in Equation 3). This result further predicts that if hot-adapted lines have evolved a means to deal with the increased cellular stress of hot temperature (35 °C), they must have done so mainly via increased resource acquisition ($M$ in Equation 3).

## Thermal adaptation in life-history traits

We assessed the predicted agricultural impact of life-history evolution by assaying three classic traits; juvenile development time, female adult body mass, and lifetime reproductive success (LRS: number of adult offspring produced), and four rate-dependent traits assayed over the first 16 h of female reproduction; early fecundity, weight loss, water loss and metabolic rate (Fig. 4). This also allowed us to evaluate if hot lines had adapted to their temperature regime by increased energy acquisition (M; approximated by adult body mass at eclosion).

To characterize trait-values at the start of experimental evolution, we assayed the three ancestors reared at 23, 29 and 35 °C. This allowed us to quantify both the direction and magnitude of adaptation during experimental evolution, as well as its subsequent agricultural impact (see further below). Temperature affected the expression of all seven life-history traits. For development time, body mass, early fecundity and LRS, there were also main effects of the geographic origin of the ancestors. Body mass, early fecundity and LRS also showed a significant interaction between geographic origin and assay temperature, indicating differences in thermal plasticity between ancestors (Fig. 4A–G, Supplementary Table 2).

To assess temperature-dependent life-history evolution we assayed the 12 evolved lines when reared in a common garden experiment including 23 °C and 35 °C. All traits except metabolic rate and weight loss showed significant differentiation between the hot and cold regime. In particular, the hot regime had indeed evolved larger adult body mass and higher reproductive output (Fig. 4A–G, Supplementary Table 3), in accordance with model predictions (Fig. 1C). All traits except metabolic rate and LRS showed a significant interaction between regime and assay temperature, demonstrating thermal adaptation sensu stricto (Fig. 4B–G, Supplementary Table 3). Several alternative estimates of laboratory fitness also showed clear signs of temperature-specific adaptation, with hot lines outperforming cold lines at 35 °C, while there were only small differences at 23 °C (Supplementary Fig. 3). In addition, for metabolic rate there was an interaction between evolution regime, assay temperature and geographic origin, indicating that thermal adaptation was contingent on founding genetic variation (Fig. 4B–E, Supplementary Table 3). Analysing all traits together using a non-parametric MANOVA[76] confirmed main effects of assay temperature ($F_{1,12} = 150.6$, $P < 0.001$), evolution regime ($F_{1,12} = 35.6$, $P < 0.001$) and the interaction between evolution regime and assay temperature ($F_{1,12} = 5.6$, $P = 0.003$).

## Thermal adaptation shapes the agricultural footprint

To predict how the observed life-history evolution may affect agricultural yields, we estimated the relationship between life-history traits and the amount of host consumption by *C. maculatus*. We reared cold, hot and ancestral lines at 23, 29 and 35 °C. At each assay temperature, we let females lay eggs in a petri-dish containing ad libitum black-eyed beans. The number of eggs laid, the number of individuals surviving to pupation, the mass of adult beetles, and the seed mass consumed, were calculated from each dish.

The hot regime consumed significantly more of the host than the cold regime ($F_{1, 5.97} = 22.8$, $P = 0.003$), demonstrating that the agricultural footprint indeed had evolved. The difference between the hot and cold regime tended to be greater at 35 °C, although the two-way interaction between regime and assay temperature was marginally non-significant ($F_{2, 11.9} = 3.25$, $P = 0.075$, Fig. 5A). To assess the importance of life-history traits in affecting food consumption, we performed analyses while sequentially including information on the number of eggs laid, juvenile survival, and body mass, so that the final model predicted the amount of host consumed as a function of the cumulative beetle mass produced (Supplementary Table 4, Supplementary Fig. 4). Adding life-history traits to the models reduced the differences between evolution regimes and in the final analysis including all traits we no longer found a significant effect of

regime ($F_{1, 5.73} = 0.60$, $P = 0.47$), demonstrating that life-history adaptation alone seems responsible for the evolved differences in host consumption. Indeed, a model containing only the main effects of eggs laid, adult body mass and juvenile survival to pupation explained 93% of all the observed variance in bean consumption across the entire experiment (Fig. 5A, Supplementary Table 4). The amount of host mass lost was 3–4 times greater than the mass of beetle produced, which is similar to what has been reported in a previous study[77], underlining the severe impact that *C. maculatus* can have on host crops.

We predicted the agricultural impact of *C. maculatus* at cold and hot temperature based on the observed life-history evolution by calculating the agricultural footprint as:

$$\varphi1 = C_m \omega M / d_1, \qquad (5)$$

where $C_m$ is the temperature-specific constant converting beetle mass produced to host mass consumed (Fig. 5A), $\omega$ is lifetime adult offspring production of a single female (Fig. 4A), M is body mass at adult emergence (Fig. 4B) and $d_1$ is mean development time in days (Fig. 4C). Hence, this measure of the footprint captures the magnitude of agricultural loss per time unit as a function of an organism's life-history strategy[78]. To assess the sensitivity of our predictions to both estimation error and choice of life-history variables we calculated a second independent estimate of the footprint, based on measurements of a different set of individuals:

$$\varphi2 = \psi C_E / d_2, \qquad (6)$$

where $\psi$ is the number of eggs laid by a single female over her first 16 h of reproduction (Fig. 4D), $C_E$ is the amount of host consumed per laid egg, and $d_2$ is the mean development time in days of the fastest developing individual in each petri-dish (both variables estimated from the host consumption experiment presented in Fig. 5A). These two metrics were highly correlated ($r = 0.88$, $t_{31} = 10.4$, $P < 0.001$, Fig. 5B) suggesting high predictive accuracy.

To provide a generalisable metric of the agricultural impact of life-history evolution under climate change, we scaled the predicted footprint for all lines relative to their respective ancestor reared at 29 °C (Fig. 5C). This metric allowed us to assess both the direct effect of temperature (i.e. thermal plasticity) on the footprint in the ancestors, and how evolution, relative to this direct effect of temperature, impacted the footprint. Cold (23 °C) and hot (35 °C) assay temperature reduced the footprint by 35-45% and 0–25% respectively, relative to benign 29 °C in the ancestors (Fig. 5C). However, evolution had a fundamental impact on how temperature affected the footprint. While cold-adapted lines showed very little difference to ancestral lines at 23 °C (Δfootprint = 7.6%; CI: 2.0 – 15%), Hot-adapted lines left almost twice as large a footprint as the ancestral lines at 35 °C (Δfootprint = 80%; CI: 70–97%). Moreover, these effects were contingent on the geographic origin of the founders. At 23 °C, cold-adapted lines from California left a slightly larger footprint compared to the ancestor (Δfootprint = 29%; CI: 13–42%) while the footprint had not evolved on the Brazil (Δfootprint = -3.1%; CI: -11 – 6.3%) and Yemen (Δfootprint = 1.0%; CI: -11 – 11%) background. At 35 °C, the footprint evolved substantially on all backgrounds, but the increase relative to the ancestor tended to be greater for Brazil (Δfootprint = 84%; CI: 64 – 110%) and California (Δfootprint = 93%; CI: 71–120%) compared to Yemen (Δfootprint = 62%; CI: 44–87%) (Fig. 5C).

These results indicate that life-history evolution can cause significant changes to the agricultural impact of insect pests over a relatively short period of time. Due to the thermodynamic effects on generation time[56], our evolved regimes had undergone different numbers of generations at the time of assaying (60 for hot vs. 45 generations for cold lines for traits reported in Fig. 4, after 5 years of

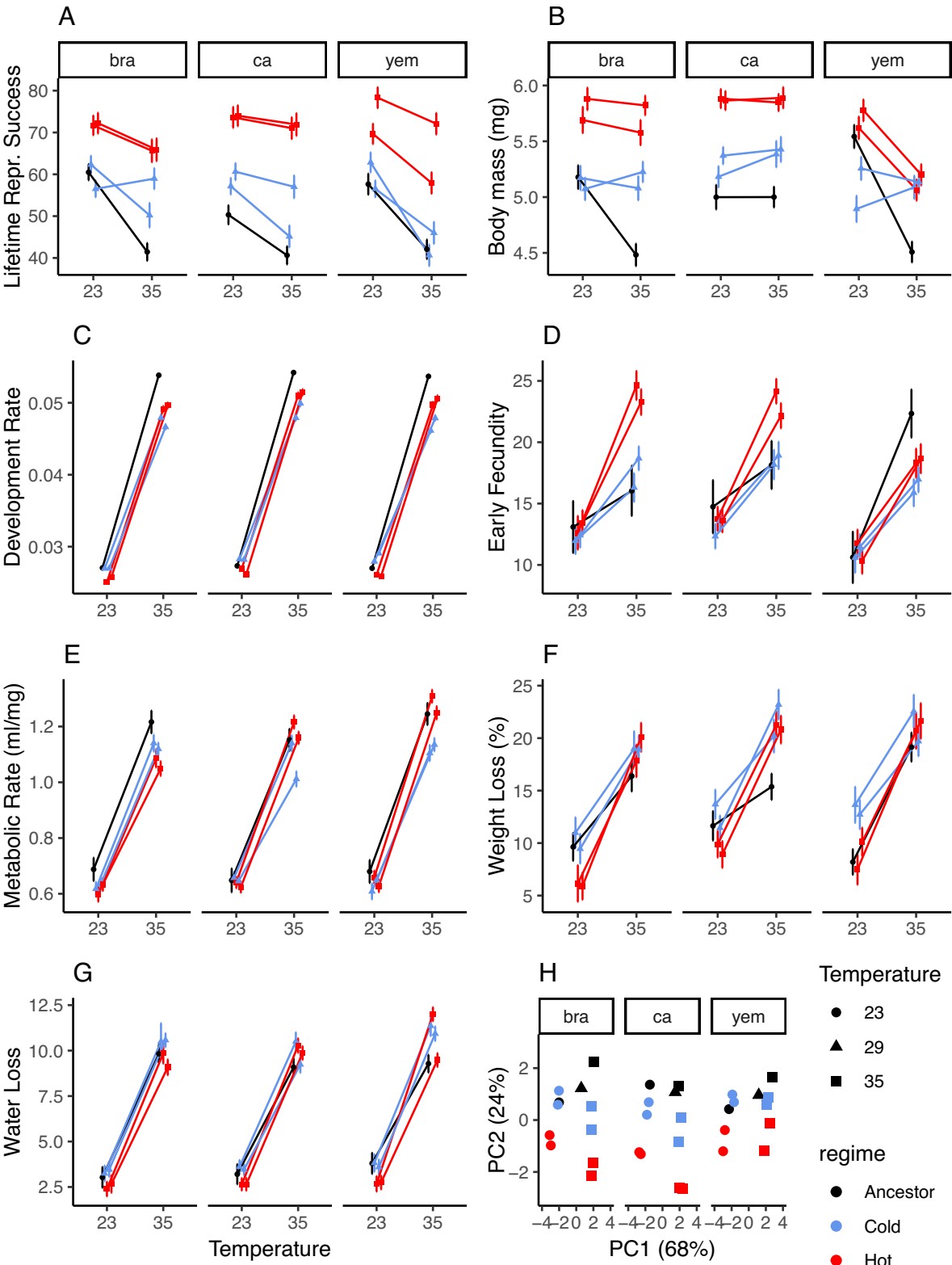

**Fig. 4 | Thermal adaptation in life-history traits.** Single traits in (**A**–**G**) for hot (red) and cold (blue) adapted lines, as well as their ancestors (black). Development time is presented as a rate (1/development time in days) for consistency with other traits. Means ± 1SE are presented for each experimental evolution line based on technical replication within each line; sample sizes for each trait is reported in Methods. **H** Ordination using principal component analysis showing that multivariate divergence occurs along both a temperature-sensitive axis (PC1) and along a general life-history axis describing overall variation in body mass and LRS (PC2), with more divergence from the ancestor in hot-adapted lines. Note that all traits presented in panels **A**–**G** except development rate were measured for females reared in groups of three.

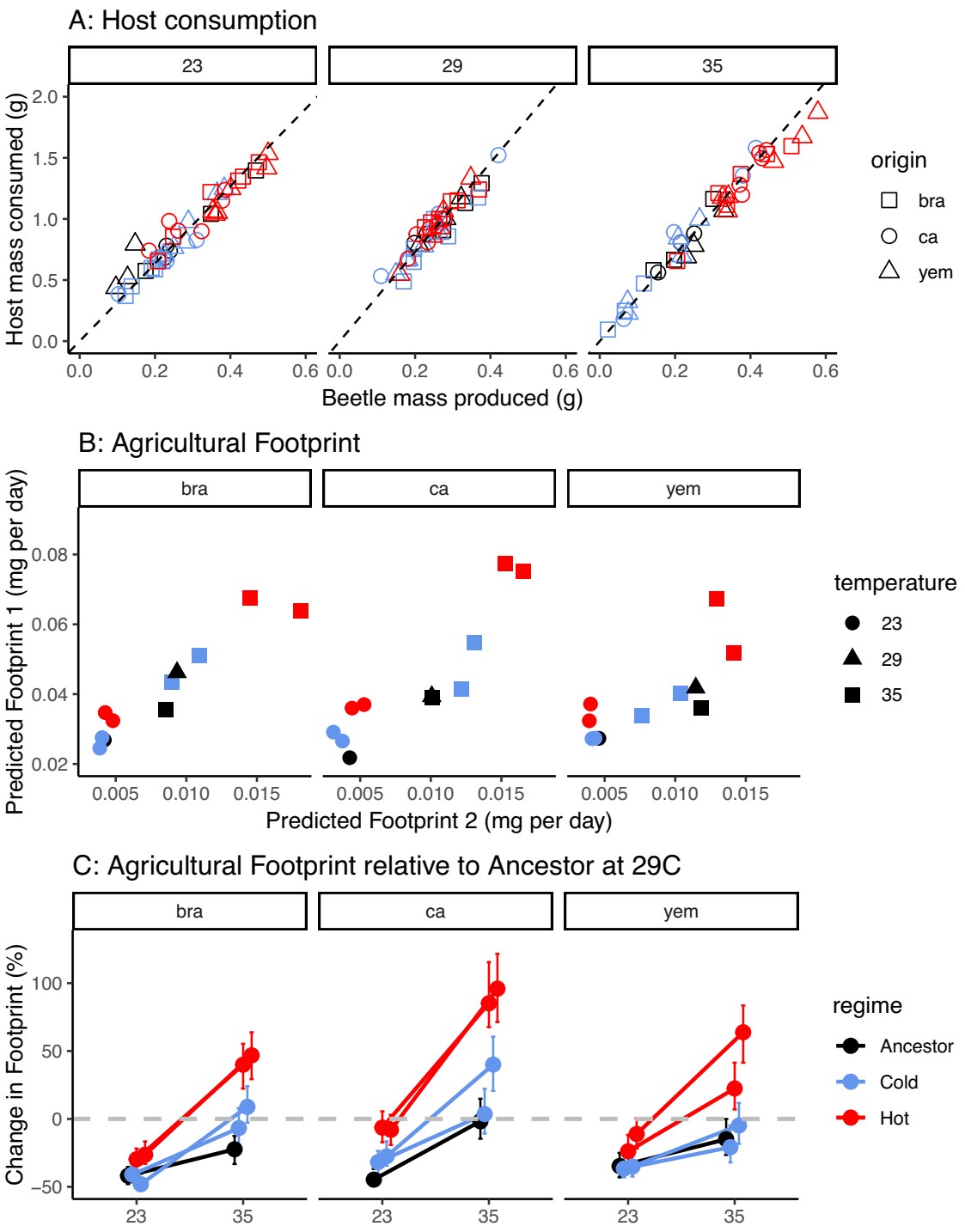

**Fig. 5 | Temperature-dependent evolution of the agricultural footprint. A** The relationship between beetle mass produced and host bean consumed. Each assay is represented by an individual datapoint. The slopes of the dashed lines correspond to the temperature-specific constant ($C_m$) giving the amount of host consumed per beetle produced (23 °C: 3.17, 29 °C: 3.66, 35 °C: 3.55). **B** The two independent metrics used to predict the agricultural footprint (see equations 5 and 6). **C** The predicted footprint of each line (using equation 5) relative to respective ancestor raised at 29 °C (grey hatched line). Means ± 95% CIs are presented for each experimental evolution line and were calculated based on equation 5 and parametric bootstraps based on standard errors for each underlying trait (see Fig. 4A–D). The effect of life-history adaptation at cold temperature is small, but life-history adaptation at hot temperature almost doubles the agricultural footprint relative to the ancestral impact.

evolution). Reported effects should correspond well with predictions for populations having continuous generations adapting at cooling and warming temperatures, respectively. However, for seasonal populations that are restricted in the number of generations per year

by other ecological factors than temperature, per-generation estimates may be more relevant. Assuming that trait-evolution proceeded at a constant (linear) pace in our experiment, the average change in the agricultural footprint due to adaptation at hot

temperature (relative to ancestral populations) would correspond to a 1.3% (1.2–1.6) increase per generation, with the estimate for cold adaptation being 0.2% (0.05-0.35).

## Discussion

Climate change is predicted to fundamentally impact growth rates and distribution ranges of ectotherms[5,30,31,42,79–81], many of which are pests on economically important crops[1,2,7,16]. Current predictions of increased agricultural impacts of pests typically rely on deterministic statistical projections of how temperature affects ectotherm metabolism and population growth, but do not incorporate evolution[13,14,78,82]. However, rapid evolutionary responses are characteristic of many pest species[1,5,7,16]. Here we therefore asked to what extent the incorporation of evolution can alter current projections of climate-induced changes in pest impact. We show three features that emphasize that future projections need to account for thermal adaptation in pest species' life-history. First, adaptation affected the agricultural footprint left by *C. maculatus*, but the effect was dependent on temperature; with little or no effect of cold adaptation while hot-adapted populations caused an almost doubling of the footprint relative to ancestors (Fig. 5). Second, these effects depended on the ancestral founder of the adapting populations, suggesting that geographic differences in segregating genetic variation in life-history traits may influence future agricultural loss under climate warming. Third, attempts to incorporate niche evolution in species' distribution models may need to do so by relatively simple means in the absence of empirical data that allow parametrisation and evaluation of more complex mechanistic models[13]. However, our study echoes concerns that such predictions may be off the mark because the mechanisms and type of traits that mediate niche evolution have a strong influence on pest impact[14,17,40,83].

Indeed, the observed temperature-dependent effect of evolution on the agricultural footprint can be understood by considering trait-specific responses. Even though cold-adapted populations did show evolved responses in traits such as weight loss and water loss, these traits are predicted to have small to no effect on the agricultural footprint. Cold populations also showed genetic responses in development rate and metabolism (the latter dependent on genetic background) that could impact agricultural loss[1,2], but these genetic changes were very modest compared to the responses observed for lifetime reproductive success, early fecundity and adult body mass, that evolved more readily in hot-adapted populations (Fig. 4). Note that all rate-dependent traits (panels A and D-G of Fig. 4) were measured in females reared in groups of three. Thus, our measures are different from estimates performed on singly isolated females, such as those quantifying resting metabolic rate, as our measures allow behavioural interactions between individuals to influence estimates of physiology.

The responses of hot-adapted populations are in line with two predictions from biophysical theory. First, warmer temperatures are predicted to ease thermodynamic constraints on metabolic rate[60], which can benefit larger size via stronger selection on fecundity[84–86]. Second, evolution at stressful temperatures may favour compensatory feeding and sequestering of resources devoted to maintaining cellular homoeostasis[5,19,20,36,37,49,70,87] (Fig. 1, Supplementary Note 1). Indeed, increased larval feeding and larger adult body mass in *C. maculatus* is associated with extended adult lifespan and greater fecundity under aphagous conditions corresponding to laboratory and grain storage environments (e.g refs. 88,89). These effects are likely magnified at hot temperatures that increase metabolism and water loss (refs. 23,87,90 and Fig. 4), and the need for upregulation of molecular chaperones to secure cellular homoeostasis (refs. 20,68,69 and Fig. 3). In accordance with this hypothesis, previous studies on *C. maculatus* also found that evolution at hot temperature increased body mass[91,92], an adaptive genetic response that paralleled the observed thermal plasticity in body mass[91]. Indeed, also for our populations used here, even though

body size is largest when beetles are raised at 23 °C, beetles reared at 35 °C are larger than when raised at benign 29 °C (Supplementary Fig. 4). To further evaluate the hypothesis that a larger size is beneficial at hot temperature in *C. maculatus*, we used the estimates of female life-history to calculate selection differentials on body size at hot and cold assay temperature. For all three estimates, selection on size was stronger at the hot temperature (Supplementary Fig. 5, Supplementary Table 5). Thus, the evolved increase in the agricultural footprint at hot temperature observed here is likely mediated by two synergistic effects of life-history adaptation, (i) increased larval feeding enabling larger size at maturity, and (ii) the positive effects of a large adult size on survival and reproduction at hot and aphagous conditions.

These observations agree broadly with the model prediction that stressful temperatures can favour increased host consumption and energy allocation to maintenance to mitigate thermal constraints on growth and reproduction (Fig. 1, Supplementary Note 1). Note, however, that this prediction does not necessarily equate to the evolution of a large size at maturity, as resources might be spent on higher metabolic expenditure and maintenance at warm temperature prior to the organism reaching adult maturity, and our simple model makes no explicit predictions about adult size at maturity (see also refs. 93,94). In line with this argument, experimental evolution at hot temperature in *D. melanogaster* fruit flies resulted in the evolution of higher foraging rates but maturation at a smaller adult size[95]. The parallel response in juvenile feeding in *C. maculatus* and *D. melanogaster* thus follows the predictions of our model, whereas the difference in effects of hot temperature on size at maturity may be explained by adult feeding (compensating for a small size at maturity) only being possible in *D. melanogaster*. Crucially, increased juvenile energy intake necessitates access to host plants. If food availability becomes limiting under climate warming, increased metabolic needs and demands on maintenance could instead severely limit thermal tolerance and population growth rates of heat exposed ectotherms[37,96]. Thus, while host abundance is already acknowledged as a key factor regulating pest outbreaks[1,7,78,97], the interdependency between heat stress and the benefits of energy acquisition may magnify the effects of host accessibility on pest outbreaks under future climate warming (Fig. 1C, D). Manipulating temperature and larval densities simultaneously and examining the outcome for both population growth and host consumption could test this prediction explicitly and further evaluate our model's predictions.

Early findings suggested that most ectotherms follow the "temperature-size rule"[98,99], stating that body size shows plastic reductions with increasing temperature. However, later results have shown that these patterns are far from general. In particular, terrestrial ectotherms, the group to which many insect pests belong, show no consistent plastic response to temperature, and genetic adaptation along latitude shows a trend towards larger body sizes at warmer latitudes[100], which is consistent with the evolved response observed here. Moreover, thermal responses in body size tend to vary even within and between closely related species[81], as for *C. maculatus*[91,101] and other Chrysomelid beetles[102]. Understanding whether taxonomic variation represents constraints rooted in thermodynamics or adaptive plasticity may offer insights into the future routes of evolution under climate warming but has proven a major challenge for the field of eco-physiological adaptation[5,19,81,93,99]. Moreover, as discussed above, the effect of heat on energy expenditure suggests that the temperature scaling of adult body mass may not follow a one-to-one relationship with that of host consumption. Controlled experimental approaches as used here may provide further clues to what extent patterns of thermal plasticity are indicative of future genetic responses to climate and how the evolution of classic life-history traits affects host consumption. Such general information could be used to increase accuracy of future projections for species where thermal plasticity can be accessed, but where evolution is hard to study directly.

In contrast to the measured life-history traits, feeding efficiency was not strongly affected by assay temperature and did not evolve in our experiment, suggesting that differences in the agricultural footprint of insect pests may be accurately predicted from life-history variables alone. Yet, our study also highlights the important role of founding genetic variation in dictating life-history adaptation. This suggests that evolution under climate change may proceed along different trajectories across a species distribution range, even when regional climates change in parallel. When such divergent evolutionary responses involve key life-history traits, as observed here, this could lead to regional differences in pest impact[1,2,83]. In addition, while our assays of offspring production and host consumption capture variation stemming from temperature effects on larval survival and male fertility, our study is focused on female life-history traits. Future studies might also want to consider the role of male life-history, fertility, and male-female mating interactions, which can be affected by temperature and have strong effects on population regulation[29,103–108], as also shown for *C. maculatus*[109–112].

The impact of insect pests on agriculture is paramount[11,78], with severe repercussions in human societies; 2.6 billion people (26% of the world's population) are currently directly dependent on agriculture for a secure living[113]. In African countries alone, the financial loss from pests was estimated to be at least 1 billion USD for 2019, with agriculture accounting for 99% of this loss in central regions[9,10]. Notably, while statistical projections that do not incorporate the potential for evolution predict that pest impact will be reduced or unaffected by climate change in these regions (e.g[2].), our results suggest that rapid pest adaptation may offset these predictions and instead result in an increased agricultural footprint. Moreover, the redistribution of species has already caused massive ramifications for the Earth's ecosystems[114], and the fear is now that future climate warming will allow insect pests to colonize new habitats[1,2,4,7,11,12]. To mitigate all these effects, management strategies need to be equipped with accurate predictive models[13,14,82,115–117], as well as an understanding of the limits to their accuracy[43,118–120]. It is our hope that our study can contribute towards this development by (i) providing general insights into the repeatability and mode of thermal adaptation in life-history of insect pests, which can inform on the general accuracy of ecological forecasting of species responses to climate change, and (ii) by emphasizing the need for incorporating a mechanistic and genetic basis of thermal niche evolution into eco-evolutionary forecasts, on which improved management decisions can be based[6,13,17,82,83,115,121,122].

## Methods

### Experimental evolution
*C. maculatus* is a pest on legume seeds and is common in tropical and sub-tropical regions[123]. Females lay eggs on the surface of seeds. After the egg hatches, the larva burrows into the seed to feed until it pupates. Adults emerge from seeds after ca. 3 weeks at standard temperatures around 29 °C and are reproductively mature almost instantly and do not require food or water to reproduce[74,75,91]. These characteristics have likely contributed to *C. maculatus* being a cosmopolitan pest that invades seed storages[12,50]. Indeed, *C. maculatus* has become a model organism for research in both evolutionary ecology and pest management because laboratory and grain storage environments are very similar.

The experimental evolution protocol exposed six lines to 35 °C (hot regime) and six lines to 23 °C (cold regime), for 85 and 130 generations, respectively (difference due to effects of temperature on generation time). The lines were derived from three ancestors that were collected from different geographic regions (Brazil, Yemen and California (USA)), with two replicate lines per thermal regime created per region. Each line was created by mating 150 females with 150 males from the respective ancestral founding population. Population size was immediately expanded to ~3000 beetles in the next generation

and beetles were split in groups of 600 to create four population replicates; two placed at hot, and two placed at cold evolution temperature per ancestor[75]. The three founders were kept at ancestral 29 °C throughout experimental evolution (Fig. 2A). A humidity of 50–55% was used throughout this study. Ancestors were reared, both before and during experimental evolution, by placing 300-400 newly emerged adults on 150 ml black-eyed beans, whereas evolution lines were reared by placing 600 adults on 250 ml black eyed beans, resulting in similar densities of adult beetles per bean. To reduce direct selection on development time, all lines were transferred to a new jar with fresh beans around the peak of emergence of adult beetles at each generation. This occurred ~3 days after the first beetle emerged for ancestors and the Hot regime, and 5 days for the Cold regime.

### Gene expression
To provide evidence for a temperature-dependent trade-off between reproduction and maintenance, we leveraged transcriptomic data from three separate experiments. The first two experiments estimated gene expression responses in female abdomens to heat shock and mating, respectively, in a separate lab population collected from Lomé, Togo[124]. The third dataset estimated expression in female abdomens in response to rearing temperature in our experimental evolution lines. All beetles were reared at standard laboratory conditions of 29 °C and 55% relative humidity unless otherwise stated.

To identify heat stress responsive genes, we compared reproductively active control and heat shocked females. Individuals were collected as virgins and kept individually. On the next day, females were heat shocked in individual 60 mm petri dishes at 55 °C for 20 min, which is stressful and reduces female fertility while still representing a thermal stress occurring at the sampling site of the population[110,111]. After a 2.5 h recovery at 29 °C and 55% rh, all females were mated to males. Females were then kept in individual 60 mm dishes until being flash frozen in liquid nitrogen 2–3 h, 6–7 h or 24–25 h after mating. The experiment was repeated on three consecutive days (blocks). Each replicate sample consisted of 5 pooled female abdomens from the same treatment, time-point and block. For sequencing, we selected a total of seven replicate samples per time-point for the control treatment and five replicate samples per time-point for the heat shock treatment. Preliminary analyses showed that there were only three differentially expressed genes at the last time-point, which therefore was removed in the final analyses (although results did not qualitatively change if including them), resulting in 24 RNA libraries for analysis.

To identify reproductive genes, we compared virgin and mated females 24 h after a single mating. We mated females to males from 8 different genetic lines. The lines had undergone 53 generations of experimental evolution under one of three mating regimes manipulating the relative strength of natural and sexual selection[111]. Accordingly, we had 8 samples of mated females, with 2 or 3 samples per male mating regime. Additionally, we had 3 samples of virgin control females. Flash freezing of all females occurred 24 h after mating, with females kept singly and allowed to lay eggs on fresh beans during the 24 h period. The final 11 RNA libraries each consisted of 12 female abdomens.

To quantify expression of reproductive and heat stress responsive genes (identified in the first two datasets) in our experimental evolution lines and their ancestors, we reared all lines at 23, 29 and 35 °C. All lines were first propagated for one generation at 29 °C to remove potential difference between evolution regimes stemming from temperature-induced (non-genetic) parental effects. Once adults emerged at respective rearing temperature, 5 males and 5 females were placed together in a 90 mm petri dish with access to fresh beans. Beetles were allowed to reproduce for 23 h (for 29 and 35 °C) or 46 h (for 23°C) after which females were flash frozen. The longer time allowed for reproduction at cold assay temperature was given to make

a more direct comparison across females from different temperatures; reproductive output (Fig. 4B), metabolism (Fig. 4E) and weight loss (Fig. 4F) is about twice as fast at 35 °C compared to 23 °C. Two petri dishes of females were pooled into one sample (i.e. 10 female abdomens pooled in total) for extractions and further analysis, resulting in one sample per line and temperature for the hot and cold regime. Two samples were taken per temperature for each of the three ancestors. This resulted in a total of 54 RNA libraries.

RNA from all samples was extracted using the Qiagen RNeasy Mini kit with beta-mercapto-Ethanol added to the lysis buffer and an on-column DNase treatment with the Qiagen RNase-free DNase kit. Tissue lysis was done in a bead mill with two stainless steel beads at 28 Hz for 90 s and RNA was eluted in two times 30–50 μl water. Some samples went through an additional clean-up step to increase RNA concentrations with the Qiagen RNeasy Mini Kit after the RNA extraction. Next generation sequencing of the samples was done at the SNP&SEQ Technology Platform in Uppsala. Libraries were prepared using the TruSeq stranded mRNA library preparation kit with polyA selection. Libraries were sequenced in one flowcell as paired-end 150 bp reads on a NovaSeq 6000 system.

Raw reads were quality checked with FastQC 0.11.9[125] and MultiQC 1.11[126] and afterwards cleaned up using Trimmomatic 0.39[127]. Adaptors were trimmed from the raw reads and ends below an average accuracy of 99% over a five base pair sliding window were clipped. Finally, reads shorter than 20 bp were discarded. Resulting read pairs were mapped to the *Callosobruchus maculatus* genome[128] using HISAT2 2.2.1[129] with default settings for stranded libraries and sensitivity set to "–very-sensitive". Number of reads per gene was then determined using HTSeq 2.0.2[130] with default settings for stranded libraries. This resulted in 10–27 M (heat stress), 13–30 M (reproduction) and 9–35 M (rearing temperature during experimental evolution) uniquely mapped reads per library going into the final downstream analyses.

Read count data was pre-processed and analysed in R 4.3.1[131] using edgeR[132,133]. For the heat shock response, only genes with at least 3 counts per million in at least 5 samples were further analysed, resulting in 10246 genes for final analysis. For the mating response, only genes with at least 1 count per million in at least 2 samples were further analysed, resulting in 11,640 genes for final analysis. The last approach was also used for the genes expressed in the evolution regimes, resulting in 11,292 genes for analysis. Counts were normalized using the 'Trimmed Mean of M-values' method via edgeR's NormLibSizes() function. Count data for the heat shock and mating response was further analysed with edgeR[132,133] and limma[134] using linear models on the normalized log2 transformed counts per million to determine differentially expressed genes. All analyses used a false discovery cut-off of 5%.

For reproductive genes, effect sizes (i.e. log-fold changes) were averaged across the three male selection regimes. However, we only considered genes that were significantly differentially expressed in all three separate contrasts between virgin females and females mated to males from the three mating regimes. For heat stress genes, effect sizes and significance was calculated by averaging across the two time-points. Significant genes were further subject to gene ontology analysis, using the R package GOstats (2.66.0)[135] and the HyperGTest function. Results were visualised using clusterProfiler (4.10)[136].

To provide a powerful, yet, unbiased estimate of temperature-dependent allocation between reproduction (mating response) and maintenance (heat stress response) in the experimental evolution lines, we focused on the 134 genes that displayed significant antagonistic differential expression patterns in response to heat shock and mating (Fig. 3B). We first multiplied the estimated log fold changes of the antagonistic genes for both mating and heat shock responses by the vector of normalized read counts of the same 134 antagonistic genes found for each of the 54 RNA libraries from the experimental evolution lines. The resulting values were then summed to compute a score for the heat stress and reproduction response separately. As expected, the scores for heat stress and reproduction, based on only the antagonistic genes, were tightly negatively correlated ($r = -0.999$) across all combinations of evolved lines and assay temperatures and fell along a singular linear trade-off axis (Supplementary Fig. 2). The response along the allocation trade-off for each sample was taken by projecting samples along this axis (quantified as the first eigenvector of the data). We repeated the same analyses using all differentially expressed genes in response to heat shock (765) and mating (1269) with qualitatively similar results (Supplementary Fig. 2, Supplementary Tables 1d, e).

## Life-history traits

We quantified thermal adaptation in female life-history by measuring three core traits: lifetime reproductive success (LRS), juvenile development time and adult body mass, and four rate-dependent traits: early fecundity, weight loss, water loss, and metabolic rate over the first 16 h of female reproduction. All life-history traits were collected at generations 45 for cold-adapted lines and 60 for hot-adapted lines, in a large common garden experiment including the two assay temperatures corresponding to the experimental evolution treatments (23 °C and 35 °C). Ancestral lines were scored in the same experimental conditions with the addition of the ancestral 29 °C assay temperature, but on a later occasion following ca. 125 generations of experimental evolution. Note that the ancestors had been kept at the ancestral conditions to which they had already adapted for >200 generations prior to the start of experimental evolution. It can therefore be assumed that the measured trait values will not have changed much due to selection during experimental evolution. Further, while we cannot completely exclude a role of genetic drift in the measured life history traits, the censuses size of 300-400 adult beetles in the ancestors corresponds to an effective population size of roughly 200[109], and the measured traits are likely under relatively strong (stabilizing) selection. Hence, we expect any effect of drift to be small and that the measured trait values should correspond reasonably well with those at the start of experimental evolution. To control for potential differences in the separate experiments on ancestors and evolved lines stemming from unknown sources, we reared an independent laboratory adapted reference population in both experiments. This indicated that differences in rearing had affected the life-history traits scored over the first 16 h of reproduction. We therefore standardized the traits scored during respirometry of the three founding ancestors by this estimated amount (adult mass: increased by 6.4%, metabolic rate: reduced by 12%; early fecundity: reduced by 18%; water loss: reduced by 15%, and weight loss: reduced by 25%) in order not to erroneously assign these differences to evolutionary divergence between ancestors and evolved lines. Note that this was done averaged across the three assay temperatures and geographic origins. Therefore, our approach to provide more accurate measures of evolutionary divergence between evolved lines and ancestors did not affect the estimated temperature-dependence of adaptation or the importance of geographic differences.

Before assays of life-history traits, non-genetic parental effects were removed by moving F0 grandparents of the assayed individuals into a common temperature of 29 °C to lay eggs. The emerging beetles in the next (parental) F1 generation were allowed to mate and lay eggs on beans provided *ad libitum*. Following 48 h of egg laying, the beans were split and assigned to one of the two (for ancestors, three) assay temperatures. The emerging adult F2 offspring were phenotyped for their life-history (Fig. 4). Newly emerged (0–48 h old) virgin females were mated to males by placing three males and females together in a petri dish over night at the assay temperature. In the following morning, the three females were weighed for their body mass and then placed together inside a glass vial filled with black eyed beans (ca. 20–25) to be measured for their metabolic rate, water loss and early

fecundity at their designated assay temperature. We note here that our goal was to measure metabolism in conditions that correspond to the most common state of adult seed beetles and most relevant ecological setting to compare the evolved populations. *C. maculatus* mate on the first day of eclosion, and females from our lines start laying eggs almost immediately at high adult densities. Further, early laid eggs are likely to be more important for individual fitness than eggs laid at older age due to the larval competition that subsequently occurs among offspring inside seeds. We thus measured metabolic rate during the first day of mated adult life for female trios. We nevertheless note that our measures of metabolism may not directly correlate with commonly used measures of resting metabolic rate from single individuals kept in isolation.

The glass vials (diameter = 2 cm, length = 4 cm) were integrated into a high-throughput respirometry system and housed inside a Sanyo MIR-153 incubator (set to an assay temperature of either 23, 29 or 35 °C, with lights turned on). Briefly, the respirometry was set up in stop-flow mode, with incoming air controlled at a flow rate of 50 ml/min by an SS-4 pump (Sable Systems, Las Vegas, NV, USA) and a Model 840 mass control valve (Sierra Instruments, Monterey, CA). $CO_2$ production and water-loss were measured for up to 23 vials on a given experimental day using a LiCor 7000 infrared $CO_2$ gas-analyser (Lincoln, NE) and a RH-300 water vapour pressure metre (Sable Systems), respectively. The system was connected via three RM-8 (eight-channel) multiplexors (Sable Systems). The first vial was left empty and served as a baseline to control for any drift of the gas analysers during each session and was measured at both the start and end of each cycle. Vials were measured over 17 cycles, each of a length of 57.5 min. The LiCor gas analyser was span calibrated using 1000ppm $NO_2$ and $CO_2$ in the morning before each trial began. Mean metabolic rate and water loss for each vial was calculated across cycles 2–17, with the readings from the first cycle discarded (as it contains human-produced water and $CO_2$). Respirometry data was retrieved using ExpeData Pro v1.5.6 (Sable Systems). After respirometry, females were weighed again to record their weight loss and beans with eggs were isolated and counted to record early fecundity. In total we followed 386 triads of females for the evolved lines and another 115 triads from their ancestors.

From the same rearing we measured egg-to-adult development time for two technical replicates per line and assay temperature, each consisting of 40–120 individuals. We calculated a mean development time per technical replicate and used this in analysis. We also collected virgin males and females and placed three males and three females together in a petri dish with ad libitum beans to record lifetime reproductive output (LRS) at each assay temperature. In total we recorded LRS for 258 couple triplets for evolved lines, and another 115 couple triplets for the ancestors. These data were complemented with additional data from both evolved and ancestral lines reared in a common garden design in two consecutive years (corresponding to generation 120/135 for ancestors, 115/130 for hot-adapted lines, and 80/90 for cold-adapted lines). In these rearings, a single male and female were put together in a petri dish with ad libitum host seeds, with otherwise the same conditions. For ancestors we scored 396 couples, and for evolved lines 789 couples, across both experimental years. LRS was analysed per female, hence we divided all offspring counts from female triads by three before analysis.

We carried out all statistical analyses using the statistical and programming software R (v. 3.6.1). All seven traits were first analysed separately using linear mixed effect models available in the lme4 package[137]. When analysing differences between hot- and cold-adapted lines, evolution regime, geographic origin and assay temperature were added as fully crossed fixed effects. Line replicate crossed with assay temperature and nested within geographic origin were added as random terms to assure correct level of replication when estimating significance of fixed terms including evolution regime. For analyses on metabolic rate, water loss and weight loss, the date of the respirometry

run and the measurement vial ID were added as additional random effects. Metabolic rate and weight loss were corrected for body mass of the measured female triad by taking an average of the weight measures before and after respirometry and adding it as a covariate in analysis. We first fitted individual slopes for the logged weight covariate (with logged $CO_2$ as response) for the hot and cold evolution regime. As these estimates were very similar (slope cold = 0.95, slope hot = 1.0) and not different statistically ($P = 0.7$), we estimated a shared slope across evolution regimes for the final model. This slope was equal to 0.98 and not significantly different from 1, as seen for other studies on *C. maculatus*[88,138,139]. We therefore chose to graphically present mass-specific metabolic rate as produced $CO_2$ in ml per minute and milligram of beetle (Fig. 4E). For water loss measured from evolved lines, measures from vial 2 (the first vial in the measuring sequence containing beetles) and measurements from three entire experimental days were discarded following an outlier analysis as these measures were magnitudes greater than other measures and clearly represented water vapour from other sources than the beetles. For LRS, experiment (triads and two separate rearings of single couples) and its interaction with assay temperature were added as additional fixed effects to account for possible block effects. The data from ancestral lines were analysed in similar fashion but without the fixed effect of evolution regime and random effect of population replicate.

*P*-values were calculated using the car package and type II sums of squares with the Kenward-Roger approximation for the degrees of freedom. To analyse evolutionary responses while taking all traits into consideration at once, we also performed a non-parametric MANOVA using residual randomization[76] with the same structure for fixed effects but where observations were based on line means, as all traits were not recorded in the same individuals (Fig. 4H).

## The agricultural footprint

We set up both evolved and ancestral lines in a common garden experiment including the 23, 29 and 35 °C assay temperature, following ~85, 120 and 125 generations of experimental evolution for cold-adapted, hot-adapted and ancestral lines, respectively. We removed parental effects by moving all lines to 29 °C two generations prior to the start of the experiment. Newly laid F2 eggs were split among the three assay temperatures and resulting virgin adults were collected. Three males and three females were placed together in petri-dishes with ad libitum *V. unguiculata* seeds for 24 h at 29°C and 35°C, and 34 h at 23 °C (as reproductive rate is slower at cold temperature; Fig. 4B). Three dishes were prepared per line and assay temperature. Note that the experiment was aimed at linking life-history variation to host consumption, and the sample size was low for detecting more fine-scaled differences between evolution regimes due to the imprecise measures of fecundity (6–9 females per line and temperature measured over 24–34 h), which were much better estimated in the previous experiments (~80 females per line and temperature across the entire adult stage).

To quantify the amount of host seed consumed by beetles, each petri dish with beans was weighed prior to the addition of beetles, to record a starting weight of each assay. After beetles had emerged and been removed, assays were placed in standard conditions for 6 weeks during which they were weighed on several occasions to record the weight of the infested beans and to check for potential inconsistencies and time-dependencies in estimates of bean consumption. However, the measures of the beans' weight loss following infestation were highly repeatable ($r^2$ = 94%) and we chose the final (fourth) measure to calculate our estimate of host consumption. To account for potential weight loss of beans that was independent of the beetle infestation, we also set up 5 assays without beetles at each temperature and measured them in the exact same way. The host consumption was then corrected based on the weight changes in these control assays; this correction was very small and only corresponded to 1–2% of the total bean weight.

For each assay, we also recorded the number of hatched eggs laid on beans (i.e. fecundity), the number of those that resulted in the adult pupating (i.e. juvenile survival), and the mean body mass of emerged adult beetles.

Differences in food consumption were first analyzed using linear mixed effect models in the lme4 package with the same structure for fixed and random terms as described for life-history traits. We excluded the ancestors from formal analysis since these had only half the samples compared to the evolution regimes, but we included them in Fig. 5A to show the direction of evolution of host consumption. We sequentially added information on fecundity, juvenile survival, and mean body mass for each assay to explore how life-history traits accounted for host consumption. We finally fitted a linear model including only the three traits (removing the fixed terms of assay temperature, evolution regime and geographic background) to estimate how much of the variation in host consumption that could be predicted by life-history alone.

Motivated by the observation that the life-history traits accounted for >93% of the observed variation in host consumption (see Fig. 5A and ref. 77), we calculated the two independent measures of the agricultural footprint (see main text) from our measured life-history traits in the main experiment, as this data was estimated with high accuracy for all lines. To provide 95% confidence limits and *P*-values, we performed parametric bootstrap on all calculations using the means and standard errors for each trait per line and assay temperature, derived from the univariate mixed models (Fig. 4A–G, Supplementary Table 3).

**Reporting summary**
Further information on research design is available in the Nature Portfolio Reporting Summary linked to this article.

## Data availability
All data on life-history traits, host consumption, along with R code for all analyses, are available at Figshare: https://doi.org/10.6084/m9.figshare.26048389. The gene expression data are deposited in NCBI with the following accessions: Mating response (PRJNA1119937 [https://www.ncbi.nlm.nih.gov/bioproject/PRJNA1119937]), Heat stress (PRJNA1120212 [https://www.ncbi.nlm.nih.gov/bioproject/PRJNA1120212]), Evolution and Thermal Plasticity (PRJNA1121056 [https://www.ncbi.nlm.nih.gov/bioproject/PRJNA1121056]).

## Code availability
The R code underlying the model predictions in Fig. 1 and Supplementary Note 1 is available in Supplementary Code 1.

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

## Acknowledgements
We thank Johanna Liljestrand-Rönn for help in the lab, and Elina Immonen and Göran Arnqvist for helpful input on our study design and results. Sequencing was performed by the SNP&SEQ Technology Platform in Uppsala. The facility is part of the National Genomics Infrastructure (NGI) Sweden and Science for Life Laboratory. The SNP&SEQ Platform is also supported by the Swedish Research Council and the Knut and Alice Wallenberg Foundation. This work was supported by grant no. 2019-05023 from the Swedish Research Council (VR), grant no. 2022-01117 from Formas, and grant no. CTS22:2101 from Carl Tryggers Stiftelse, to D.B.

## Author contributions
E.B., C.G.T., M.M., E.C. and J.B. planned, coordinated and performed the collection of all phenotypic data. M.K. and E.B. planned and performed the collection of gene expression data. A.R. and A.F.H. analyzed all gene expression data. D.B. designed the study, analyzed the phenotypic data, constructed the model, and wrote the manuscript. All authors read and commented on the first manuscript draft and approved the submission.

## Funding

## Competing interests
The authors declare no competing interests.
