## [Transparent Peer Review file · Nature Communications]

Life-history adaptation under climate warming magnifies the agricultural footprint of a cosmopolitan insect pest

Corresponding Author: Dr David Berger

Version 0:

Reviewer comments:

Reviewer #1

(Remarks to the Author)

The authors examine how evolutionary responses to warming climates will influence agricultural loss (amount of crop destroyed from insect foraging) from an insect pest species (seed beetle). They present an impressive dataset where they experimentally evolved beetles at two different temperatures and conducted a range of genetic and trait assays to examine evolutionary responses to climate change, and how adaptation of pest species might impact agriculture.

While I think that accounting for evolutionary responses to climate change are very important, especially in multiple traits, overall, I found the manuscript to be a bit confusing and disjointed. The introduction has a heavy focus on climate change and how there is an important need to understand impacts of evolution at warmer temperatures. But there is very limited discussion in the introduction about the metabolic theory of ecology and additional thermodynamic hypotheses that the authors delve into quite heavily in the results. More background information is required to explain what hypotheses they are testing and why.

I was also confused about the hypotheses surrounding plasticity. The first mention of plasticity is in the results and then it is discussed in the discussion. But there is no background on what aspects of plasticity they are testing and why.

I have some concerns about how the authors tested metabolic rate. It is very difficult to make inferences on how metabolic rate evolves when groups of individuals are tested within the same chamber. In addition, methods on how metabolic rate was tested need additional clarification. There is no information on how many gas analysers were used and if the system was actually calibrated or not (span gas calibration and flow rates).

While the amount of data generated is truly impressive, and I imagine the study would have taken a lot of people, funding, and effort, I don't think the story is cohesive yet.

Abstract

Overall comments from abstract: I found the abstract vague. I didn't really know what the authors were testing in their study. It would be helpful if they could clarify a few statements.

Line 33: what life history theory?

Line 36: what kind of traits?

Line 38: Change in what kind of traits? What kind of agricultural footprint?

Line 42: not sure if can generalize to all insects from only one species.

Introduction

Generally, I liked the introduction. It is indeed important to understand evolutionary responses to climate change while also

examining range shifts. However, I felt like the introduction suffered from the same vagueness as the abstract. To understand what the study is about, and the importance of different traits in the face of climate change, I think it would be helpful if the authors could be a bit more explicit about what traits they are testing and why. They hint that they will be testing some traits linked to metabolic rate, but don't specifically state what life history traits they are testing.

Missing introduction heading.

Line 60 – 62: direct thermodynamic effects do make MR increase, but the metabolic cold adaptation hypothesis predicts that cold adapted organisms actually have higher metabolic rates than warm adapted organisms.

66 – 72: trade-offs don't always seem apparent, for example in many cases, jack-of-all-trades are also 'masters-of-all' (rather than masters-of-none'), suggesting that there are not always trade offs between performance and thermal generalism.

74: Any specific type of life history evolution? Starting the paragraph with however is confusing me.

84: what traits are you looking at?

88: what is the agricultural footprint? How much the beetles are expected to eat?

87 – 93: vague, what kind of life history traits?

Methods

The metabolic rate methods require clarification. Specifically, its suboptimal to test multiple individuals in a chamber at once if you really want to understand how climate evolution impacts individual metabolic rate.

427: how many individuals per line to start off with?

451: is that long enough? Why not a bit longer at a slightly lower temperature?

454: were the hours until freezing randomised across the three days?

454: why the different time until freeze periods?

489: why only some samples?

574: I assume that the number of generations differed so that traits could be collected at the same time. But the number of generations could also impact evolutionary responses.

579: why did you use flow stop mode instead of just flow through? What kind of gas analysers were you using? Licore?

575: why did you test the MR of three individuals at a time?

576: why were they placed into the metabolic rate chamber with food? Digestion can impact metabolic rate.

580: how many analysers were you using? Were you using multiplexors?

582: did you use a gillibrator (or equivalent to make sure flow rates were accurate)? Did you use span gasses to calibrate your analysers?

611: chamber/analyser should also be included as a random effect as analysers can be out by different degrees (especially if span gases were not used to estimate CO₂ concentration correction factors).

631: instead of assay temperature do you mean exposure temperature?

Results

General comments: I feel like a lot of the information in the first two paragraphs would be better suited for the introduction, on why particular traits are being examined and what kinds of hypotheses are being tested.

117: This scaling exponent is highly debatable, and it has been shown that the allometric relationship between body mass and metabolic rate itself can evolve.

164: predictions on how traits and temperature-specific responses influence agricultural impact are not clear in the introduction.

239: finding no differences in metabolic rate could be due to measuring multiple individuals at a time, and potentially data could be off if the gas analysers and flow rates were not calibrated.

Discussion

321: interesting that there was no significant cold adaptation, often rates of cold tolerance are supposed to evolve faster than rates of heat tolerance (e.g. Bennett et al. 2021) <https://www.nature.com/articles/s41467-021-21263-8>

335: It looks like lifetime reproductive success declines in the hot evolved groups. Thus, wouldn't reduced reproductive output be indicative of a smaller agricultural footprint?

338: thermodynamic effects of temperature make metabolic rate increase, which usually means that organisms require more food, and therefore selection is generally thought to favour smaller body sizes (need less food to support organism with high MR)? That's part of why Bergman's rule suggests that larger animals occur in colder locations. Also, your data show that body mass decreased at hotter temperatures, which supports Bergmann's rule. So I'm confused about your sentence "First, warmer temperatures are predicted to ease thermodynamic constraints on metabolic rate⁵², which can increase selection for larger size via effects on fecundity".

357: the results and discussion focus heavily on allocation, but predictions and hypotheses on allocation are not really discussed in detail in the introduction. Thus, the introduction and discussion seem disconnected.

Reviewer #2

(Remarks to the Author)

Here the authors evaluate the result of experimental evolution in *C. maculatus* using three geographic populations under three temperatures. One possible scenario is an up-regulation of molecular chaperones that maintain cellular homeostasis, with the downstream consequence that the beetles increase resource intake and/or divert resources away from reproduction. Using thermodynamic equations, the authors find the greatest agricultural impact at temperatures slightly above those that maximize growth rate. They examine previous data showing differential expression in response to heat shock and between virgin and reproducing females. They find 137 genes common to both data sets. Interestingly, most of the overlapping genes are differentially regulated between datasets, suggesting a trade-off between reproduction and heat stress response. The authors then compute a score based on these gene expression differences to evaluate potential trade-offs in their experimentally-evolved lines, finding that beetles from all regimes have an allocation towards cellular maintenance at the highest temperature. They then assessed seven life-history traits in the evolved lines to assess the agricultural impact of their evolution experiment, finding that beetles evolving under the hottest temperature had larger body mass and reproductive output. Beetles evolving under the hottest temperature also consumed more of the host bean, suggesting that the agricultural damage was increased.

This is an intriguing study that will be of interest to general readers. The manuscript was well-written and the methods were well articulated, which will enable reproducibility. I have the following comments.

Major comments:

1. Methods, lines 529-533, Results, lines 200-210, Suppl. Table 2c, and Figure 3: the authors propose to calculate an estimate of temperature-dependent allocation between the reproduction and heat stress response. To do this, they multiply "...the estimated log fold changes of the antagonistic genes for both mating and heat shock responses by the vector of normalized read counts of the same 134 antagonistic genes found for each of the 54 libraries from the experimental evolution lines. The resulting values were then summed to compute a score for the heat stress and reproduction response separately." The log fold changes are presumably coming from the Lome, Togo results previously described, though this is not entirely clear. Using this metric they then find that beetles reared at 35 degrees have a "strong change in allocation towards cellular maintenance and away from reproduction compared to beetles reared at the other two temperatures...", with no significant differences observed among evolution regimes. I am wondering whether the authors might see differences among the evolved populations if they instead split their metric by log-fold changes for each type of allocation, i.e., evaluate the metric separately for the high fold change reproduction genes (upper left quadrant in 3B) and then for the high fold change heat shock genes (lower right quadrant in 3B). Or perhaps I have misunderstood this analysis and that is what the authors did.

2. Results, the authors seem to have missed an opportunity to identify differences in gene expression across selection

regimes.

Minor Comments

1. Supplemental Table 2b, it is not clear what P-value correction was used in the GO analysis. Raw P-values appear to be listed in the table.
2. Methods, It isn't clear what the light cycle conditions were for the experiment.
3. Results, lines 553-554, the authors state that because the ancestors have been maintained at the same conditions, "it can therefore be assumed that the measured trait values correspond well with the trait values at the start of experimental evolution." This claim seems dubious as random genetic drift could influence these parameters over time, unless the population size was very large.
4. This manuscript focused on female beetles. Some acknowledgement of differences with males could be articulated in the Discussion.
5. Methods, line 494, "...NovaSeq 600 system" appears to be a typo.

Reviewer #3

(Remarks to the Author)

General comments

I enjoyed reading this study. It is excellent to see the combination of experimental evolution, modelling, molecular biology, life history, and physiology. I have only a few specific comments that I think should be integrated in a revision, which I hope that you find useful (note that I have restricted my comments to the areas in which I feel qualified to provide useful comments: primarily the physiology, and the life history).

Specific comments

L29 and elsewhere (e.g. L48-49, 59) – ectothermic would be a better choice than "cold-blooded". Ectothermic species primarily use environmental sources of heat to thermoregulate, whereas endothermic species primarily use endogenous heat. Either could be considered "cold-blooded" dependent on the body temperature that they maintain and the frame of reference, and so "cold-blooded" is not an informative term.

L30. Could you provide a little more information about the "dire consequences"? If climate change was forcing pests to retreat from arable land, then the consequences might not be dire, for example, so it is best to be explicit.

L121-123. Given that the conclusions would seem likely to be quite sensitive to evolution in A or Ea in equation (1), I think it would be helpful to provide some further information about your statement that Ea evolves more slowly than A.

L151-152. What exactly do you mean by "reducing thermodynamic constraints on reaction rates"? – I think that you mean that reaction rates increase with temperature, such that the constraint is that reaction rates are constrained to be low at low temperatures? But there are examples, such as Hochachka and Somero's classic example of LDH activity being nearly invariant with temperature (your reference 20), whereby this does not apply.

L546. The use of mass-specific metabolic rates is appropriate for comparison among groups that differ in size if, and only if, metabolic rate scales in direct proportion to metabolic rate. Such isometric scaling is exceedingly rare, and metabolic scaling exponents are typically around 0.75, as you note in the text. In your statistical analysis you have included body mass as a covariate in the analysis (L612-614), which is appropriate, however this introduces a part-whole correlation in the data because mass now appears both in the response and the predictor. This can lead to spurious correlations. As such, it would be more appropriate to express the metabolic rate data as whole-organism metabolic rate, and analyse with body mass as a covariate.

Version 1:

Reviewer comments:

Reviewer #2

(Remarks to the Author)

The authors have thoroughly addressed all of my concerns.

(Remarks on code availability)

The authors provide the code as supplemental data. It appears to be a valid R code.

Reviewer #3

(Remarks to the Author)

Thank you for your thorough responses to my queries, especially for the clarification regarding the statistical analysis (now clarified this in the Methods on L713-721) and for your willingness to make modifications to the text to ensure that your meaning is clear. I have no further suggestions for revision.

(Remarks on code availability)

Response to reviewers:

Reviewer #1 (Remarks to the Author):

The authors examine how evolutionary responses to warming climates will influence agricultural loss (amount of crop destroyed from insect foraging) from an insect pest species (seed beetle). They present an impressive dataset where they experimentally evolved beetles at two different temperatures and conducted a range of genetic and trait assays to examine evolutionary responses to climate change, and how adaptation of pest species might impact agriculture.

Thank you for acknowledging the amount of work behind our study.

While I think that accounting for evolutionary responses to climate change are very important, especially in multiple traits, overall, I found the manuscript to be a bit confusing and disjointed. The introduction has a heavy focus on climate change and how there is an important need to understand impacts of evolution at warmer temperatures. But there is very limited discussion in the introduction about the metabolic theory of ecology and additional thermodynamic hypotheses that the authors delve into quite heavily in the results. More background information is required to explain what hypotheses they are testing and why.

We have added more background regarding thermal adaptation in the Introduction (**L56-70**). We have also added to the topic of acquisition and allocation trade-offs (**L72-87**). We prefer to keep the Introduction relatively short and instead expand on some points as we go along, including thermodynamics and the metabolic theory of ecology that we introduce in the Results as part of the model (which we think works as a more technical and extended introduction). We hope these edits will please the reviewer.

I was also confused about the hypotheses surrounding plasticity. The first mention of plasticity is in the results and then it is discussed in the discussion. But there is no background on what aspects of plasticity they are testing and why.

We have made some brief clarifications about the role of plasticity in response to temperature (e.g. direct effects of temperature on rates of reproduction and mortality) and the consequences for agricultural impact in the Introduction (**L56-65**).

I have some concerns about how the authors tested metabolic rate. It is very difficult to make inferences on how metabolic rate evolves when groups of individuals are tested within the same chamber. In addition, methods on how metabolic rate was tested need additional clarification. There is no information on how many gas analysers were used and if the system was actually calibrated or not (span gas calibration and flow rates).

We agree that it is hard to draw any conclusions about the evolution of resting metabolic rate. However, our aim was to describe metabolic rates in the setting at which these beetles evolve. Females get mated on their first day of adult life and then spend the rest of it laying eggs, which is done at densities like those used during our measurements. Further, female seed beetles lay most of their eggs early in life. Hence, we measured metabolic rate of groups of females when performing their most important activity – laying eggs, during their first day as mated adults. For our purposes, we believe that this measure is relevant for understanding thermal adaptation in life history and its mechanistic basis in seed beetles. We have clarified this and mention the caveat of extrapolating to individual resting metabolic rates, in the Methods (**L660-668**).

We span-calibrated the gas analyser (LiCor 7000) in the mornings before runs. Flow rates were not calibrated on a daily basis but of course recorded. We have added a more detailed description of the respirometry methods (**L670-688**) which we hope will please the reviewer.

While the amount of data generated is truly impressive, and I imagine the study would have taken a lot of people, funding, and effort, I don't think the story is cohesive yet.

Thank you for acknowledging the amount of work going into this study and for the constructive suggestions. We have done our best to clarify our hypotheses, results, and the conclusions we draw from them.

Abstract

Overall comments from abstract: I found the abstract vague. I didn't really know what the authors were testing in their study. It would be helpful if they could clarify a few statements.

We note that the abstract in the submitted version was 199 words which we have needed to shorten to 150 words. Thus, while we appreciate the reviewer's comment, we do not really see a way to accommodate this, given the journal's length requirements. Our main has thus been to give the take-home message and entice the readers to keep on reading.

Line 33: what life history theory?

The theory stating that variation in key life history traits is governed by allocation trade-offs and acquisition trade-offs. We have tried to explain this better but acknowledge that there is little room to explain this in greater detail and we hope readers will accept this.

Line 36: what kind of traits?

see above

Line 38: Change in what kind of traits? What kind of agricultural footprint?

see above

Line 42: not sure if can generalize to all insects from only one species.

We agree that more studies on a wide variety of species are needed, we have therefore replaced “will” with “can” in the sentence. Nevertheless, our model makes general predictions, which were aligned with the results we found. We have expanded slightly on our previous discussion of the generality of the model predictions and the need for further verification using other species to confirm their generality (**L419-452**).

Introduction

Generally, I liked the introduction. It is indeed important to understand evolutionary responses to climate change while also examining range shifts. However, I felt like the introduction suffered from the same vagueness as the abstract. To understand what the study is about, and the importance of different traits in the face of climate change, I think it would be helpful if the authors could be a bit more explicit about what traits they are testing and why. They hint that they will be testing some traits linked to metabolic rate, but don't specifically state what life history traits they are testing.

Thank you for this concrete and constructive comment. We have expanded on the Introduction as suggested. We have added information on which traits that can be affected and are important for pest species impact on agriculture on **L57-62, 77-85, 102-104 & 107-112**.

Missing introduction heading.

Added. Thank you.

Line 60 – 62: direct thermodynamic effects do make MR increase, but the metabolic cold adaptation hypothesis predicts that cold adapted organisms actually have higher metabolic rates than warm adapted organisms.

We agree with this sentiment and did expect to find this effect of cold-adaptation, but it seems not to be the case in general for metabolic rate in our study (Fig. 4E). Note also that we found local adaptation of development rate with respect to temperature (Fig. 4C). We have expanded slightly on the previous comment on this in the Discussion (**L388-394**).

66 – 72: trade-offs don't always seem apparent, for example in many cases, jack-of-all-trades are also 'masters-of-all' (rather than masters-of-none'), suggesting that there are not always trade offs between performance and thermal generalism.

Trade-offs are often not found because of complex biology and/or methodological aspects (e.g. measuring performance in a simple lab environment relative to the wild, or not measuring all relevant traits). This is in fact one of the main points we make by raising the possibility that responses to heat or cold stress might not entail adaptation in the

thermodynamic properties of enzymes, but via compensatory energy intake (**L68-87**). We hope the revisions to the Introduction have improved the description of this complexity.

74: Any specific type of life history evolution? Starting the paragraph with however is confusing me.

We agree. We have re-written this part as a consequence of the other changes to the Introduction in response to your comments.

84: what traits are you looking at?

We have moved some of this information from the Results to here (now **L107-112**) so that we now mention that we measured development rate, body mass, metabolic rate and reproduction.

88: what is the agricultural footprint? How much the beetles are expected to eat?

Yes, this is what we meant. To clarify, we have changed this so that it instead reads "Here we explored how adaptation to climate change in life-history traits of insect pests affect their population growth rates and host consumption" (**L99-100**).

87 – 93: vague, what kind of life history traits?

We have added more detail so that the main results are hopefully clear now (**L107-112**).

Methods

The metabolic rate methods require clarification. Specifically, its suboptimal to test multiple individuals in a chamber at once if you really want to understand how climate evolution impacts individual metabolic rate.

See answer above. Our goal was slightly different – we do believe that the most relevant measure of metabolism corresponds to that taken at the time when the beetles spend most of their adult life and fitness-enhancing activity. We have however, as noted above, made some clarifications regarding this issue on **L660-668** and **L670-685**.

427: how many individuals per line to start off with?

Lines from a given origin were started by first mating 150 females with 150 males from each source (ancestral) population and was then immediately expanded to a population size of ~3000 beetles in the next generation, just before experimental evolution started, when beetles were split in groups of 600 into each replicate. We have added this information on **L501-505**.

451: is that long enough? Why not a bit longer at a slightly lower temperature?

We have done a few studies on heat-shock in *C. maculatus* and protocols like this one are both ecologically relevant and induces visible effects on offspring production – see citations in the manuscript (Baur et al. 2022 Functional Ecology, and Baur et al. 2023, Evolution Letters).

454: were the hours until freezing randomised across the three days?

We see that this was written in a slightly confusing way. All census times were done for each of the three blocks of the experiment (i.e. repeated on three occasions). We have added a clarification on **L528-534**.

454: why the different time until freeze periods?

We wanted to study the effect of time since we were not sure of when the heat stress response would be most obvious.

489: why only some samples?

This was done to increase RNA concentrations of those samples in order to fulfil the criteria for sequencing at the platform. We have clarified this.

574: I assume that the number of generations differed so that traits could be collected at the same time. But the number of generations could also impact evolutionary responses.

Indeed, but we felt it appropriate to provide evidence for thermal adaptation using a common garden design. We think a reassuring result is that, for traits measured both at the main life-history phenotyping (generations 40 & 60 - Fig. 4) and those measured again to estimate host consumption (generations 80 & 120 – Fig 5A) (development rate, body mass and fecundity) we see very similar and consistent differences between the hot and cold regime. Nevertheless, to address this comment, we have also provided discussion and per-generation estimates of the change in footprint due to evolution at cold versus hot temperature, on **L353-364**.

579: why did you use flow stop mode instead of just flow through? What kind of gas analysers were you using? Licore?

Yes, Li-Cor 7000 (information added). We used stop-flow because the beetles are small and stop flow is more accurate for our study organism.

575: why did you test the MR of three individuals at a time?

see previous answer.

576: why were they placed into the metabolic rate chamber with food? Digestion can impact metabolic rate.

To stimulate egg laying (also see previous answers). Note that adult seed beetles do not feed on the beans, these are egg-laying substrate and food for the larvae (the only hatch a week later).

580: how many analysers were you using? Were you using multiplexors?

Yes, one CO₂ gas-analyser with three multiplexors. We hope that the added detail to the description of metabolic rate measurements will please the reviewer.

582: did you use a gilibrator (or equivalent to make sure flow rates were accurate)? Did you use span gasses to calibrate your analysers?

We did not calibrate flow rates but did perform span calibration of CO₂ each morning of the experiments. We have added this information (**L670-685**).

611: chamber/analyser should also be included as a random effect as analysers can be out by different degrees (especially if span gases were not used to estimate CO₂ concentration correction factors).

The chambers/vials were indeed added as random effects in analyses, but we failed to include this information in the previous version. We have now clarified this in the manuscript. (we only used a single analyser so this cannot be added as a factor in the analysis). Note that all R code for model fitting is available at Figshare.

631: instead of assay temperature do you mean exposure temperature?

The assays were performed at these temperatures, to which the beetles were exposed. We prefer to keep it like this to keep it consistent throughout the manuscript.

Results

General comments: I feel like a lot of the information in the first two paragraphs would be better suited for the introduction, on why particular traits are being examined and what kinds of hypotheses are being tested.

We hope that now, with the edited Introduction, the reviewer is more content with the bridge between Introduction and Methods.

117: This scaling exponent is highly debatable, and it has been shown that the allometric relationship between body mass and metabolic rate itself can evolve.

While we agree with this sentiment, it did not evolve in our experiment - comparing the hot and cold populations (log-log slope for cold = 0.95; hot = 1.0, p-value for interaction = 0.7). Hence, we kept it as a constant in our model. We note that the qualitative conclusions from our model are not dependent on what that constant is set to within

reasonable limits (e.g. a positive number between 0.5-1.5). A higher coefficient leads to stronger selection on energy acquisition, but in similar ways in populations adapting to hot and cold environments. Note that R code for the model is provided where it is possible to change the slope to see its effect on allocation and acquisition decisions.

164: predictions on how traits and temperature-specific responses influence agricultural impact are not clear in the introduction.

We hope that the reviewer is more content with the edits we made to the Introduction. Also note that we did not make predictions for all 7 traits and focused on energy acquisition (larval feeding) and adult reproduction and acknowledge that some of our findings are more of explorative in nature (e.g. changes in water loss).

239: finding no differences in metabolic rate could be due to measuring multiple individuals at a time, and potentially data could be off if the gas analysers and flow rates were not calibrated.

We did find effects of adaptation on metabolic rate, but these effects differed between genetic backgrounds (see Fig. 4E and supplementary tables). We agree with the reviewer's comment (and previous comment)- we had expected some genetic compensation of the effect of cold temperature on metabolism to evolve so that cold adapted populations would have had higher metabolic rates all else equal, but it is not what we found when measuring metabolism during female egg laying. (Note, however, that we did find consistent thermal adaptation in development rates). We refer to **L390-394** and **L665-668** where we comment on this.

Discussion

321: interesting that there was no significant cold adaptation, often rates of cold tolerance are supposed to evolve faster than rates of heat tolerance (e.g. Bennett et al. 2021) <https://www.nature.com/articles/s41467-021-21263-8>

Yes, we think that part of the reason for this is that 23C is cold, but not acute stressful for the beetles, as opposed to studies on critical thermal minima (i.e. the studies collated by Bennett et al 2021). Notably, we did find repeated evolution of slightly faster development rates of cold populations at cold temperatures across all three backgrounds, in line with expectations of thermal compensation effects. However, hot adapted populations developed faster at hot temperature. We tried not to select directly on development time in our experimental evolution design, so we think these results are consequences of local adaptation of juvenile growth in response to resource competition among larvae. In any case, these effects were not very big in magnitude and did not affect the agricultural footprint very much (as stated in the Discussion, **L388-394**).

335: It looks like lifetime reproductive success declines in the hot evolved groups. Thus, wouldn't reduced reproductive output be indicative of a smaller agricultural footprint?

35C causes a decline in reproductive success in all populations, relative to 23C, because 35C is stressful (plastic effect), but this decline is only slight in hot-evolved populations, (effect of genetic adaptation). The hot-evolved lines are also bigger and more fecund, hence their large footprint at hot temperature relative to ancestors and cold-evolved populations. Note also that reproductive rate (used to estimate our second version of the footprint) is much higher at 35C, especially for hot-adapted populations.

338: thermodynamic effects of temperature make metabolic rate increase, which usually means that organisms require more food, and therefore selection is generally thought to favour smaller body sizes (need less food to support organism with high MR)? That's part of why Bergman's rule suggests that larger animals occur in colder locations.

Yes, but note that for terrestrial insects the trend along latitude is actually opposite to what Bergmann's rule predicts (Horne et al. 2015 Ecology Letters – cited in Discussion; see **L435-440**). We believe that there is some confusion about body mass versus energy acquisition – our model predicts the latter, but we discuss the former as a correlate measure of food intake in our beetles, which are capital breeders and acquire all resources for adult reproduction in the juvenile stage (we have clarified on **L202-204**). However, as we discuss (**L419-433**), and the reviewer points out, size does not have to increase with increased food intake at hot temperature due to the high metabolic expenditure. Thus, food intake may increase at hot temperature without a change in size or even reduced final size. We probably could have done a better job of explaining this and we tried to clarify this further in the Results by being more precise in our description of the model. In doing so we changed notation to more clearly distinguish [body mass = m], as modelled by the metabolic theory of ecology, and [larval energy acquisition = M], which is what we optimize in our model. We would like to thank the reviewer for this comment that has helped us improve the clarity of our model description.

Also, your data show that body mass decreased at hotter temperatures, which supports Bergmann's rule.

Size is smaller when we rear beetles at 35C versus 23C. However, beetles reared at 35C are LARGER than beetles reared at ancestral 29C. We have added a clarification on **L407-409** and included a supplementary figure showing this pattern from the experiment that reared all lines and ancestors at a common garden quantifying the food consumption (the results presented in Fig. 5A). Here is first the supplementary figure when beetles were raised on black eyed beans (as in the rest of this study) and also a figure from a parallel experiment where we reared the beetles on adzuki beans:

Black eyed beans (now included in Supplementary 5):

Figure Redacted

This to us indicates that the increase in size at the hottest temperature (both plastically relative to 29C, and genetically by the evolution of large size in hot-adapted lines) is adaptive and we reason that it so because of the capital breeding strategy of aphagous adult seed beetles, that acquire all their adult resources in the larval stage (clarified on **L202-204**). At 35C, large adults do a lot better as they need more water and fat to survive the hot temperature, discussed on **L387-417**.

So I'm confused about your sentence "First, warmer temperatures are predicted to ease thermodynamic constraints on metabolic rate⁵², which can increase selection for larger size via effects on fecundity".

We hope that the answers above clarify the matter. The sentence above refers particularly to work showing that hot temperatures provide more opportunity for insects adapted to warm climates to realize their reproductive potential (see e.g. Berger et al. 2008, Functional Ecology) and that on a macroecological scale it seems that species with thermal optima at hotter temperatures seem to have higher maximal reproductive rates (e.g. Frazier et al. 2006 American Naturalist). (Both cited in the manuscript). We have modified the sentence slightly to improve clarity.

357: the results and discussion focus heavily on allocation, but predictions and hypotheses on allocation are not really discussed in detail in the introduction. Thus, the introduction and discussion seem disconnected.

With our edits to the Introduction together with the model laid out in the Results, we hope that there is now a better connection.

Thank you for your comments that has allowed us to clarify a range of matters and helped us to produce a more cohesive manuscript.

Reviewer #2 (Remarks to the Author):

Here the authors evaluate the result of experimental evolution in *C. maculatus* using three geographic populations under three temperatures. One possible scenario is an up-regulation of molecular chaperones that maintain cellular homeostasis, with the downstream consequence that the beetles increase resource intake and/or divert resources away from reproduction. Using thermodynamic equations, the authors find the greatest agricultural impact at temperatures slightly above those that maximize growth rate. They examine previous data showing differential expression in response to heat shock and between virgin and reproducing females. They find 137 genes common to both data sets. Interestingly, most of the overlapping genes are differentially regulated between datasets, suggesting a trade-off between reproduction and heat stress response. The authors then compute a score based on these gene expression differences to evaluate potential trade-offs in their experimentally-evolved lines, finding that beetles from all regimes have an allocation towards cellular maintenance at the highest temperature. They then assessed seven life-history traits in the evolved lines to assess the agricultural impact of their evolution experiment, finding that beetles evolving under the hottest temperature had larger body mass and reproductive output. Beetles evolving under the hottest temperature also consumed more of the host bean, suggesting that the agricultural damage was increased.

This is an intriguing study that will be of interest to general readers. The manuscript was well-written and the methods were well articulated, which will enable reproducibility. I have the following comments.

We thank the reviewer for their very accurate summary and positive evaluation of our study.

Major comments:

1. Methods, lines 529-533, Results, lines 200-210, Suppl. Table 2c, and Figure 3: the authors propose to calculate an estimate of temperature-dependent allocation between the reproduction and heat stress response. To do this, they multiply "...the estimated log fold changes of the antagonistic genes for both mating and heat shock responses by the vector of normalized read counts of the same 134 antagonistic genes found for each of the 54 libraries from the experimental evolution lines. The resulting values were then summed to compute a score for the heat stress and reproduction response separately." The log fold changes are presumably coming from the Lome, Togo results previously described, though this is not entirely clear.

It is correct that the data used to find "reproductive" and "heat stress" genes come from a population collected from Togo, but the data is not published previously. We have now uploaded all gene expression data and provided reviewer tokens in case access is wanted.

Using this metric they then find that beetles reared at 35 degrees have a "strong change in allocation towards cellular maintenance and away from reproduction compared to beetles reared at the other two temperatures...", with no significant differences observed among evolution regimes. I am wondering whether the authors might see differences among the evolved populations if they instead split their metric by log-fold changes for each type of allocation, i.e., evaluate the metric separately for the high fold change reproduction genes (upper left quadrant in 3B) and then for the high fold change heat shock genes (lower right quadrant in 3B). Or perhaps I have misunderstood this analysis and that is what the authors did.

This is an interesting idea - we tried what the reviewer suggested. We have incorporated the new results into Supplementary 2 and on **L248-257** of the main text. Unsurprisingly, the results are almost exactly the same when analyzing the 113 overlapping genes upregulated to heat stress. However, when analyzing the 21 overlapping genes upregulated to mating, we find a statistically significant increase in allocation to reproduction in ancestors relative to both the hot and cold regime, suggesting that some aspect of allocation to reproduction versus heat tolerance has changed during experimental evolution (Supplementary Fig. S2A). Nevertheless, since this response only was based on 21 genes (out of the total 1269 mating response genes), we still think that the main patterns in the data suggest that resource allocation has remained relatively unchanged compared to resource acquisition (i.e. body size evolution).

2. Results, the authors seem to have missed an opportunity to identify differences in gene expression across selection regimes.

This was intentional – we choose this strategy to focus the manuscript on the particular question at hand (i.e. testing the model assumptions and the general idea that thermal

adaptation in life history has one basis at the molecular level in the trade-off between investment in chaperones (stress response) versus investment in reproduction. We think this was necessary given the already quite complex nature of our study but we do indeed plan to investigate the molecular basis of thermal adaptation across these lines further in the future by combining the RNAseq data with DNA poolseq data.

Minor Comments

1. Supplemental Table 2b, it is not clear what P-value correction was used in the GO analysis. Raw P-values appear to be listed in the table.

We have updated the table with both raw and *fd*r corrected (at $\alpha = 0.05$) p-values.

2. Methods, It isn't clear what the light cycle conditions were for the experiment.

Assuming the reviewer is referring to the experiments on gene expression in particular, we have added this info on **L521-522**: "All beetles were reared at standard laboratory conditions of 29°C and 55% relative humidity unless otherwise stated."

3. Results, lines 553-554, the authors state that because the ancestors have been maintained at the same conditions, "it can therefore be assumed that the measured trait values correspond well with the trait values at the start of experimental evolution." This claim seems dubious as random genetic drift could influence these parameters over time, unless the population size was very large.

It is true that we cannot completely exclude genetic drift, but we note that the relative differences between hot, cold and ancestral lines in traits scored at the large phenotyping experiment (Fig. 4) and the host consumption experiment (Fig. 5A) were very similar, which we think is reassuring. The census size of the ancestors of 300-400 beetles per generation corresponds to an effective population size of ca. 200 (Martinossi-Alilbert et al. 2019 Evolutionary Applications), so we do not expect genetic drift to have played a very large role in divergence over the time frame of our experiment, given that the traits we measure are likely to be under strong stabilizing selection in the ancestors. We have added this explanation on **L631-636**.

Additionally, note that we used replicate lines in all comparisons including hot and cold populations, and assessed significant differences by including this level of replication as random effects in all models testing divergence, which excludes the possibility that drift is responsible for general differences between experimental evolution regimes.

4. This manuscript focused on female beetles. Some acknowledgement of differences with males could be articulated in the Discussion.

This is a very good idea. We have done so on **L461-466**.

5. Methods, line 494, "...NovaSeq 600 system" appears to be a typo.

Yes, thank you – changed to 6000.

Many thanks for the constructive suggestions.

Reviewer #3 (Remarks to the Author):

General comments

I enjoyed reading this study. It is excellent to see the combination of experimental evolution, modelling, molecular biology, life history, and physiology. I have only a few specific comments that I think should be integrated in a revision, which I hope that you find useful (note that I have restricted my comments to the areas in which I feel qualified to provide useful comments: primarily the physiology, and the life history).

We thank the reviewer for their positive evaluation of our study. We would like to add here that it is nice to see the acknowledgement that the combination of different approaches is a strength of our study.

Specific comments

L29 and elsewhere (e.g. L48-49, 59) – ectothermic would be a better choice than “cold-blooded”. Ectothermic species primarily use environmental sources of heat to thermoregulate, whereas endothermic species primarily use endogenous heat. Either could be considered “cold-blooded” dependent on the body temperature that they maintain and the frame of reference, and so “cold-blooded” is not an informative term. Thank you – we have changed it to “ectotherm/ic” throughout.

L30. Could you provide a little more information about the “dire consequences”? If climate change was forcing pests to retreat from arable land, then the consequences might not be dire, for example, so it is best to be explicit. We have very little room in the Abstract to do so (which needed to be shortened from 199 to 150 words). We have however inserted a “potentially” before “dire”, so make room for other scenarios like you point out.

L121-123. Given that the conclusions would seem likely to be quiet sensitive to evolution in A or Ea in equation (1), I think it would be helpful to provide some further information about your statement that Ea evolves more slowly than A. Thank you for this comment that has allowed us to clarify our approach – we realize that we were not completely accurate in our previous explanation for why we modelled thermal adaptation via energy acquisition and allocation. Our main point was that the temperature-sensitivity of enzymes (i.e. change in protein sequence that changes the thermal stability or catalytic efficiency of an enzyme; e.g. Dill et al. 2011 PNAS) seem to evolve more slowly than compensatory responses (i.e. upregulation of heat shock proteins to confer thermal tolerance, or upregulation of ribosomal proteins to facilitate protein translation and high growth rate at cold temperature, while the protein sequence of the enzyme itself remains unchanged). This is why we modelled thermal adaptation via life-history allocation and acquisition trade-offs. We have changed the paragraph to

clarify this, and we agree with the reviewer that the two modes of adaptation outlined above do not necessarily have to reflect evolution of E_a versus A (for example, compensatory responses involving increased production of ribosomal protein could be assigned as affecting E_a based on empirical measurements).

We modelled the compensatory adaptation via effects on A as this seemed both biologically motivated and was more practical; a proportional change in A leads to a proportional change in temperature-dependent reproduction and mortality rates. We have clarified this further on **L164-169** and cite a few studies as examples of the debate around thermal adaptation via A vs. E_a (e.g. Gillooly et al. vs. Clarke et al. 2006).

As we point out on **L175-179**, whether and how strong compensatory responses will be, depend on the assumptions about the stressfulness of acute hot and cold temperatures, which can be modelled in a range of ways (including assuming that compensatory adaptation affects E_a). However, as we here focused on effects of climate warming and our beetles were not reared at acute cold temperatures, we did not model lethal effects of cold temperature. We have clarified this on **L183-186** (see also the two last paragraphs of Supplementary 1).

In the process of making these clarifications, we also took the opportunity to arrange the equations and change notation slightly in the main text so that they would correspond directly to that used for the example presented in Figure 1 and Supplementary 1. We hope that with these edits our approach is better explained and motivated.

L151-152. What exactly do you mean by "reducing thermodynamic constraints on reaction rates"? – I think that you mean that reaction rates increase with temperature, such that the constraint is that reaction rates are constrained to be low at low temperatures? But there are examples, such as Hochachka and Somero's classic example of LDH activity being nearly invariant with temperature (your reference 20), whereby this does not apply.

This is what we meant and we have tried to rephrase this to make this clearer. The sentence now reads: "First, and as pointed out previously², warm temperatures increase growth rates and agricultural impact via thermodynamic effects on enzyme reaction rates." We agree that there are exceptions to the rule and that some reactions take place readily at cold temperature, but we assume the reviewer still agrees that cold temperature constrains enzyme reactions more generally. We are open to suggestions for a better way to phrase this if the reviewer is not happy with this edit.

L546. The use of mass-specific metabolic rates is appropriate for comparison among groups that differ in size if, and only if, metabolic rate scales in direct proportion to metabolic rate. Such isometric scaling is exceedingly rare, and metabolic scaling exponents are typically around 0.75, as you note in the text. In your statistical analysis you have included body mass as a covariate in the analysis (L612-614), which is appropriate, however this introduces a part-whole correlation in the data because mass now appears both in the response and the predictor. This can lead to spurious

correlations. As such, it would be more appropriate to express the metabolic rate data as whole-organism metabolic rate, and analyse with body mass as a covariate.

We think there has been a misunderstanding here due to how we explained how the trait was measured (which was a bit sloppy on our behalf). We labelled it “mass-specific metabolism” in the beginning of the Methods, however, this is how we presented it graphically, but we did in fact analyse whole-organism metabolic rate with mass as a covariate.

This was our procedure. We first estimated evolution regime-specific mass-CO₂ relationships. As these were (statistically) invariable (log(CO₂) on log(mass); cold slope = 0.95; hot slope = 1.0, p-value for interaction = 0.7), we then estimated a common slope across regimes. This slope was not statistically different from 1 (mean = 0.98, SE = 0.07). Hence, for presentation of all results, we then plotted mass-specific CO₂ (ml CO₂/min/mg; Fig. 4E). We have clarified this in the Methods on **L713-721**.

Thank you for the constructive feedback and the chance to improve our manuscript.